# Grounded in Reality: Learning and Deploying Proactive LLM from Offline Logs

## Abstract

Large Language Models (LLMs) excel as passive responders, but teaching them to be proactive, goal-oriented partners—a critical capability in high-stakes domains—remains a major challenge. Current paradigms either myopically optimize single-turn attributes or rely on brittle, high-cost user simulators, creating a persistent "reality gap". To bridge this gap, we introduce `Learn-to-Ask`, a general, simulator-free framework for learning and deploying proactive dialogue agents *directly from offline expert data*, bypassing the need to model complex user dynamics. Our key insight is to reframe the offline policy learning problem by leveraging the **observed future** of each expert trajectory. This allows us to infer a dense, turn-by-turn reward signal grounded in the expert's revealed strategy, decomposing the intractable long-horizon problem into a series of supervised learning tasks, and training a policy to output a structured (`action`, `state_assessment`) tuple, governing both **what to ask** and, crucially, **when to stop**. To ensure reward fidelity, our Automated Grader Calibration pipeline systematically purges noise from the LLM-based reward model with minimal human supervision. Empirically, we demonstrate the efficacy of `Learn-to-Ask` in a real-world medical dataset, using LLMs of varying sizes up to 32B. Our approach culminates in the successful deployment of LLMs into a live, large-scale online AI service. In rigorous in-house evaluations, our model was launched and achieved performance even superior to human experts, proving our framework's ability to translate offline data into tangible, real-world impact. We hope this work provides a practical and economically viable blueprint for transforming passive LLMs into proactive, goal-oriented LLM applications.

## 1 Introduction

Across industries such as healthcare, law, and finance, numerous goal-oriented conversations take place every day between human experts and their clients (Wang et al., 2025; Yang et al., 2023). This vast corpus of dialogue data represents a largely untapped goldmine, containing implicit expert-driven strategies for navigating complex, information-seeking scenarios. While organizations possess these valuable data assets, Large Language Models (LLMs) are seldom trained to harness them effectively. Instead, their default behavior remains largely passive, limiting their potential as truly collaborative and proactive partners. In high-stakes domains, this passivity is a critical failure – an intelligent LLM application should not merely answer questions but proactively form a policy to gather information and drive the conversation towards a designated goal.

Two main paradigms have emerged to instill such proactivity, yet both struggle with a significant "reality gap". The first, **attribute-based alignment**, decomposes proactivity into single-turn qualities like clarity or relevance, often training on synthetic preference data (Li et al., 2025b). While useful for polishing individual questions, this approach is fundamentally myopic. It optimizes for local attributes and fails to learn a coherent, sequential *policy* that accounts for temporal dependencies in a conversation. Crucially, it provides no principled mechanism for deciding **when to stop**, a decision vital for efficiency and user experience. The second direction, **simulation-based optimization**, ambitiously targets long-horizon rewards using a user simulator (Wu et al., 2025). However, for open-ended, expert-level domains, creating a high-fidelity simulator is notoriously difficult, computationally prohibitive, and suffers from a combinatorial explosion of states. Policies

optimized in a synthetic world often fail to generalize to the unpredictable nature of real human interactions, leaving the reality gap unbridged.

In this work, we ask a fundamental question: *Can we learn an effective, long-horizon questioning policy directly from offline expert data, thereby bypassing the need for a simulator and bridging the reality gap?*

We answer in the affirmative by proposing `Learn-to-Ask`, a novel and general framework for learning proactive dialogue policies from real-world conversational logs. Our core insight is to avoid simulation entirely by leveraging the rich, sequential structure of existing expert trajectories. We decompose the intractable long-horizon Reinforcement Learning (RL) problem into a sequence of tractable, single-turn learning tasks. At each turn, the agent's immediate goal is extracted from the *observed future* of the current conversation, allowing us to infer reward signals that are grounded in what a real expert actually did in the future, and not limited to the immediate next step. This enables us to train a policy that learns a structured output (`action, state_assessment`), addressing both **what to ask** and **when to stop** with a Micro-Reward to measure the question utility and a Macro-Reward to assess the conversational progress.

The efficacy of our framework, `Learn-to-Ask`, is demonstrated through a two-pronged validation. First, in offline experiments on `RealMedConv`, a real-world medical dialogue dataset, our method transforms passive LLMs into strategic agents. For example, `Qwen2.5-7B-Instruct` trained with our framework more than **tripled** its ability to ask perfectly targeted questions and learned to correctly terminate conversations with over 92% accuracy. More importantly, we bridge the "reality gap" in practice: a `Learn-to-Ask`-trained model was deployed in a live, large-scale medical AI service. It not only functioned robustly but achieved task-success rates exceeding those of human experts, providing powerful evidence that our offline learning paradigm directly translates to superior real-world performance. Our contributions are threefold:

- **A Simulator-Free Policy Learning Framework:** We propose `Learn-to-Ask`, a novel framework that learns a complete, sequential questioning policy—including a stopping condition—directly from offline expert logs. This provides a grounded, data-driven, and economically viable alternative to brittle user simulators.

- **Hindsight-based Reward Inference:** We introduce a method to infer dense, turn-by-turn rewards by using the *observed future* of expert trajectories. This is coupled with an **Automated Grader Calibration** pipeline that ensures reward fidelity with minimal human oversight, systematically mitigating oracle noise.

- **Demonstrated Real-World Impact:** We validate our framework not only via offline experiments but also report on the successful deployment of a `Learn-to-Ask`-trained agent in a large-scale commercial service. The agent achieved super-human performance on key business metrics, demonstrating a practical blueprint for translating offline data into real-world value.

## 2 RELATED WORKS

Instilling proactivity in LLMs has evolved from simple prompting (Deng et al., 2023b; Zhao & Dou, 2024) to fine-tuning on single-turn attributes using preference optimization like DPO (Li et al., 2025b; Rafailov et al., 2023). While effective for local properties (e.g., clarity), these methods are myopic and fail to learn a long-horizon, stateful policy that includes a crucial stopping condition.

To address sequential decision-making, another line of work employs reinforcement learning (RL) in simulated user environments (Wu et al., 2025). The primary drawback is the "reality gap": policies optimized in a synthetic world often fail in real human interactions, as building a high-fidelity simulator for complex, open-ended domains is notoriously difficult (Hao et al., 2024).

Our work carves a distinct path by learning a sequential policy *directly from offline expert data*, eliminating the need for a simulator. It is philosophically aligned with offline RL from human data (Shani et al., 2024; Zhou et al., 2024), but our core contribution lies in the *reward inference methodology*. We reframe the problem by using the **observed future** of each trajectory to define a dense, turn-by-turn supervisory signal, a principle inspired by Hindsight Experience Replay (HER) (Andrychowicz et al., 2017) but fundamentally adapted for learning a complete dialogue policy in

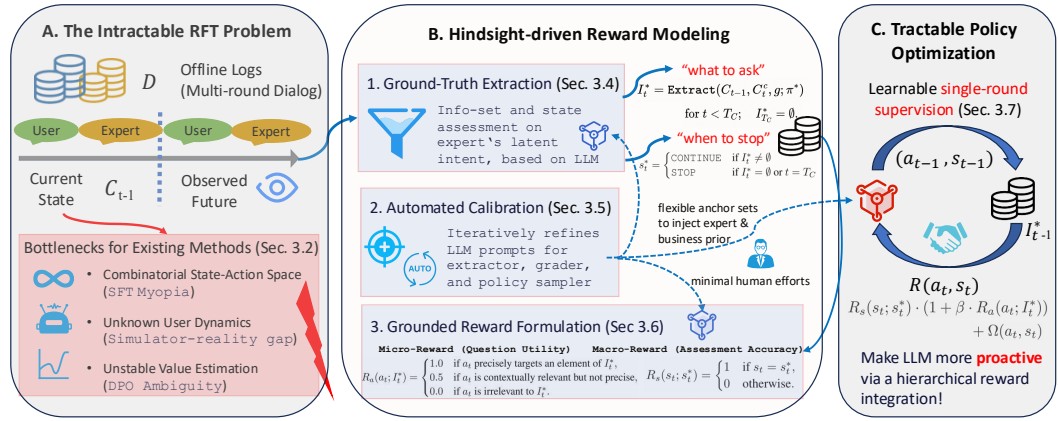

Figure 1: The overview of the proposed `Learn-to-Ask` framework. which transforms the intractable offline RL problem into a sequence of tractable supervised learning tasks.

a high-dimensional language space. More Detailed comparisons for related works are presented in Appendix B.

# 3 METHODOLOGY: THE LEARN-TO-ASK FRAMEWORK

## 3.1 PROBLEM FORMULATION: PROACTIVE DIALOGUE AS OFFLINE RL

We formulate the task of proactive, goal-oriented dialogue as a sequential decision-making problem. The agent's objective is to learn a policy, $\pi$, from a static, offline dataset of expert-led conversations, $\mathcal{D} = \{\tau_1, \tau_2, \ldots, \tau_N\}$. Each trajectory $\tau \in \mathcal{D}$ represents a complete conversation, $\tau = (u_0, x_1, u_1, \ldots, x_{T-1}, u_{T-1})$, where $u_t$ is the user's utterance and $x_t$ is the agent's utterance at turn $t$. At each turn $t$, the policy $\pi$ observes the conversation history up to that point, $C_{t-1} = (u_0, x_1, \ldots, u_{t-1})$, and generates a structured utterance tuple $x_t = (a_t, s_t)$.

Here a natural language question $a_t$ aimed at gathering new information, and a discrete state assessment $s_t \in \{\text{CONTINUE}, \text{STOP}\}$ indicating whether the agent believes the conversational goal has been met. Thus, the policy is defined as $\pi(a_t, s_t | C_{t-1})$. The learned policy should mimic the expert's strategy to complete the underlying task (e.g., medical diagnosis) effectively and efficiently. This problem can be formally modeled as learning from offline data in a Markov Decision Process (MDP) containing the following key components. (1) **State**: The conversation history $C_{t-1}$. (2) **Action**: The agent's structured utterance $(a_t, s_t)$. (3) **Transition Dynamics** ($P$): The unknown user response dynamics, which govern the state transition $P(C_t | C_{t-1}, a_t)$, where the next state $C_t$ is formed by appending the agent's question $a_t$ and the user's subsequent utterance $u_t$ to the history $C_{t-1}$. (4) **Reward Function** ($R$): The unknown reward function that implicitly guided the expert's actions. The central challenges, which we address in our methodology, are that we operate in an offline setting (we cannot query $P$) and we must infer the reward function $R$ directly from the expert trajectories in $\mathcal{D}$.

## 3.2 MOTIVATION: BEYOND MYOPIC IMITATION

Expert-led conversations are not rigid scripts but flexible traversals of an underlying information space to achieve a goal. For example, two doctors diagnosing the same patient may ask questions in different orders, but they aim to cover a similar set of critical information points. This strategic flexibility is a hallmark of expertise.

Conceptually, a goal-oriented conversation can be viewed as traversing an implicit information graph to cover a set of critical nodes. From this perspective, the limitations of prior methods become clear: Supervised Fine-Tuning (SFT) myopically learns a single path, failing to generalize to alternative valid strategies. Preference-based methods like DPO face ambiguity, as preferences are path-dependent and can yield conflicting signals when aggregated across a diverse dataset of expert trajectories. A detailed discussion and formalization is provided in Appendix C.1.

To capture a long-term strategy, we adopt the offline RL framework. However, this introduces its own well-known challenges, namely the "reality gap" from the lack of a user simulator and the instability of offline value estimation. A detailed exposition of these challenges in the context of dialogue is provided in Appendix C.2.

### 3.3 Overview: Objective Decomposition via Hindsight

To sidestep the challenges of standard offline RL, we introduce a novel objective decomposition inspired by Hindsight Learning (Andrychowicz et al., 2017). Our core idea is to reframe the intractable sequential decision problem into a sequence of tractable, single-step supervised learning tasks. As illustrated in Fig. 1, this is achieved by leveraging the **observed future of each real trajectory as a grounded oracle**.

Specifically, instead of estimating a long-horizon value, for each turn $t$, our Hindsight-driven Reward Pipeline (Part B in Fig. 1) analyzes the future conversation segment $C_t^c = C \setminus C_{t-1}$ to extract a ground-truth tuple $(I_t^*, s_t^*)$. This tuple represents: (1) $I_t^*$: The **target information set** that the expert went on to collect; and (2) $s_t^*$: The expert's implicit **stopping decision** (CONTINUE or STOP).

This process effectively transforms the original, difficult offline RL problem (Part A) into a dataset of '(state, hindsight-objective)' pairs. Consequently, we can employ stable policy optimization methods (Part C) where the goal is to train a policy $\pi(a_t, s_t | C_{t-1})$ that aligns with this hindsight-derived objective. This decomposition grounds the entire learning process in demonstrated expert strategy, teaching the policy both **what to ask** (to cover $I_t^*$) and **when to stop** (to match $s_t^*$). The subsequent sections will now detail each component of this pipeline, from ground-truth extraction (Sec. 3.4) to policy optimization (Sec. 3.7).

### 3.4 Ground Truth Extraction from Observed Trajectories

For each turn $t$ in a successful dialogue $C$ (i.e., achieved the designated goal $g$ by the end), we extract a ground truth tuple $(I_t^*, s_t^*)$ from the future context $C_t^c$. This process is guided by a powerful LLM, $\pi^*$, which acts as a *noisy oracle* for interpreting the expert's latent intent.

**Micro-Goal $I_t^*$ (Target Information Set).** This represents the set of goal-relevant information that the expert sought and obtained in the subsequent turns $C_t^c$. We define this as the "information delta" that the expert successfully closed. To extract this, we employ a powerful LLM, $\pi^*$, as an information extractor. Specifically, for each turn $t$, we prompt $\pi^*$ with the overall goal $g$, the current context $C_{t-1}$, and the future conversation $C_t^c$. The prompt instructs the LLM to identify and list only the critical new pieces of information present in the user's responses within $C_t^c$ that were not already available in $C_{t-1}$. We present the seed prompt in Appendix G and describe how to automatically refine it later in Sec. 3.5.

This structured extraction, governed by $\pi^*$, yields the target information set for turn $t$:

$$I_t^* = \texttt{Extract}(C_{t-1}, C_t^c, g; \pi^*) \quad \text{for } t < T_C; \quad I_{T_C}^* = \emptyset. \tag{1}$$

This process ensures that our micro-goal is grounded in the actual information-gathering path taken by a human expert. A crucial action in this stage is avoiding the extraction of overly generic or context-independent information, as such information could be a potential cause of reward hacking. For example, in a diagnostic conversation, physicians may commonly inquire about the pregnancy status before making a medication decision; including such information in the ground truth may result in a trained LLM to ask such a question with high probability across contexts. More implementation details can be found in Appendix F.2.

**Macro-Goal $s_t^*$ (Target Situation Assessment).** This is the ideal action (CONTINUE or STOP) at turn $t$. It reflects the expert's implicit decision. We infer this based on whether there was still critical information to be gathered:

$$s_t^* = \begin{cases} \texttt{CONTINUE} & \text{if } I_t^* \neq \emptyset \text{ and } t < T_C, \\ \texttt{STOP} & \text{if } I_t^* = \emptyset \text{ or } t = T_C. \end{cases} \tag{2}$$

This learns an expert-aligned stopping policy directly from data, a component absent in attribute-focused methods.

## 3.5 AUTOMATED PROMPT CALIBRATION

Our "learning from the future" paradigm relies on LLMs to perform three critical functions: ground-truth extraction, reward grading, and policy sampling. The behavior of these LLMs is dictated by natural language prompts, making their alignment with true expert intent a first-order concern. An uncalibrated prompt can introduce systemic bias, teaching the policy to chase phantom goals or misinterpreting its own actions.

To ensure our entire framework is robustly **grounded in reality**, we introduce **Auto-Prompt**, a unified pipeline to automatically calibrate all three prompts using minimal human supervision. This process creates a verifiable chain of fidelity from data interpretation to policy optimization:

1. **Grounding the Objective:** The **Extractor Prompt** is optimized to align its output $I_t^*$ with a small set of human-verified information goals ('anchor set'). This ensures the policy learns to pursue what a human expert would actually deem critical, preventing objective drift. We measure this alignment via F1-score, treating it as a semantic entity recognition task.
2. **Grounding the Learning Signal:** The **Grader Prompt** is refined to ensure its reward scores mimic human judgment. Its prompt is optimized to minimize the Mean Squared Error (MSE) against a small set of human-assigned quality scores, ensuring the reward function is a faithful proxy for expert-level assessment.
3. **Grounding the Exploration:** The **Policy Sampler Prompt** used during RFT is calibrated to generate a candidate action space that is both diverse and high-quality. The prompt is selected to maximize the average reward of the sampled candidates, making the policy search process more efficient and effective.

The core mechanism of Auto-Prompt is an iterative search (see Appendix E) that uses an LLM to propose prompt variations and scores them against the human-curated anchor sets. A key feature of this design is its **flexibility**; these small anchor sets can be easily updated to inject new business priorities or correct for model biases observed in production, enabling continuous, targeted improvement of the entire system without large-scale relabeling efforts.

## 3.6 GROUNDED REWARD FORMULATION

With the calibrated reward model and the extracted ground truth $(I_t^*, s_t^*)$, we can now score any candidate generation $(a_t, s_t)$ produced by our policy. Our reward function is designed to be grounded in the observable outcomes of the expert's dialogue path, rather than relying on abstract, subjective criteria. The final reward is a composition of two heads, reflecting our decomposed objective.

**Micro-Reward (Question Utility).** This component, $R_a$, measures how effectively the generated question $a_t$ targets the necessary information $I_t^*$ that the expert deemed critical to collect next. Instead of a simple binary preference, which loses significant information, we employ a graded scoring system that our calibrated grader $R_\phi$ outputs. This provides a more nuanced learning signal:

$$R_a(a_t; I_t^*) = \begin{cases} 1.0 & \text{if } a_t \text{ precisely targets an element of } I_t^*, \\ 0.5 & \text{if } a_t \text{ is contextually relevant but not precise,} \\ 0.0 & \text{if } a_t \text{ is irrelevant to } I_t^*. \end{cases} \tag{3}$$

This graded structure is crucial. The intermediate score of 0.5 helps mitigate the sparse reward problem common in dialogue tasks by crediting partially correct attempts, while the high score of 1.0 incentivizes the model to learn the kind of precision exhibited by experts. This is a significant advantage over methods that rely on pairwise preferences (e.g., DPO), which cannot differentiate between *good* and *excellent* actions with the same granularity.

**Macro-Reward (Assessment Accuracy).** This component, $R_s$, evaluates the correctness of the agent's decision to continue or stop, $s_t$, against the expert's implicit decision, $s_t^*$. This is a straightforward but critical binary reward:

$$R_s(s_t; s_t^*) = \begin{cases} 1 & \text{if } s_t = s_t^*, \\ 0 & \text{otherwise.} \end{cases} \tag{4}$$

**Reward Integration.** A key aspect of a successful policy is prioritizing the correct high-level decision (when to stop) over the low-level action (what to ask). An excellent question is worthless if asked at the wrong time (e.g., after all information has been gathered). To enforce this hierarchy, we use a multiplicative fusion function that makes the entire reward contingent on the correctness of the macro-decision:

$$R(a_t, s_t) = R_s(s_t; s_t^*) \cdot (1 + \beta \cdot R_a(a_t; I_t^*)) + \Omega(a_t, s_t). \tag{5}$$

The +1 term is added to ensure that $R_s$ is addressed even for $R_a = 0$. $\Omega(\cdot)$ is a flexible reward or penalty term to regulate the output (e.g., format and length). Its precise definition used in our experiments is in Appendix F.3. The $\beta > 0$ term is a tunable knob balancing the preference for generating good questions and making an aggressive decision, and we set $\beta = 2$ by default for all experiments. However, it is worth noting that finding an ideal $\beta$ is a non-trivial task, as it is affected by many factors, including the base model and design of reward functions. This multiplicative formulation acts as a hierarchical gate: the reward for asking a good question ($R_a$) is only granted if the strategic decision to continue is correct ($R_s = 1$). This enforces a lexicographical preference for the macro-decision, preventing the agent from receiving credit for good questions asked at the wrong time (e.g., after the goal is met). In Sec. 4.2, we will empirically compare different fusion functions and the choice of $\beta$.

### 3.7 Policy Optimization via Reinforcement Finetuning

With a structured dataset derived from real logs and a well-defined, grounded reward function, we are now equipped to train our policy. We frame this as an offline reinforcement learning problem. The dataset for training consists of tuples $\langle C_{t-1}, a_t, s_t, R(a_t, s_t) \rangle$, where $(a_t, s_t)$ are sampled responses to the context $C_{t-1}$, and $R$ is their calculated reward.

As a result, our method can be applied to extensive offline RFT algorithms without ad-hoc modifications. In our experiments, we mainly study Group Relative Policy Optimization (GRPO) (Shao et al., 2024). Unlike methods like PPO that require a separate critic model to estimate advantages, GRPO estimates advantage directly and efficiently from a group of sampled responses. This group optimization nature also utilizes the advantage of our method in exploring possible question spaces. Moreover, its group-wise advantage estimation also naturally handles the graded, non-binary nature of our rewards, as the normalization process dynamically adjusts the learning signal based on the quality distribution of sampled responses, helping to navigate the nuances of expert-level conversation. This makes it more adaptive, stable, and less complex to implement, a benefit for real-world deployment pipelines.

## 4 Offline Evaluation

### 4.1 Setups

Our experiments are conducted on `Qwen2.5-7B/32B-Instruct` models (Yang et al., 2024). The core of our evaluation is the `RealMedConv` dataset [1], which contains $2,000$ real-world pharmacist-patient diagnostic dialogues ($1,600$ for training, $400$ for evaluation). Each dialogue is segmented into turn-wise '(context, hindsight_objective)' tuples, where the objective $(I_t^*, s_t^*)$ is extracted from the observed future of the conversation as described in Sec. 3.4. The powerful `Qwen2.5-32B-Instruct` is used as the backbone for our info-extractor and reward grader. Full implementation details, including data preprocessing and training configurations, are in Appendix F.

**Baselines and Ablations.** We compare our method against the following baselines: (1) *Direct Prompting:* The base model guided by a carefully engineered zero-shot prompt. (2) *Behavioral Cloning (SFT):* Standard supervised fine-tuning to directly imitate the expert's next utterance $(a_t, s_t^*)$. (3) *Direct Preference Optimization (DPO):* We form preference pairs where the expert's response is 'chosen' and a base model's generation (which is irrelevant to any information in the context) is 'rejected' (Rafailov et al., 2023), testing if learning a simple preference for expert actions is sufficient. To validate our design choices, we conduct ablations by removing the micro-reward (*w/o $R_a$*), removing the macro-reward (*w/o $R_s$*), and replacing our hierarchical fusion with simple

---

[1] https://huggingface.co/datasets/datajuicer/RealMedConv

reward summation (*Sum*). Besides GRPO, we also evaluate `Learn-to-Ask` with other advanced RL algorithms such as CISPO (Chen et al., 2025a) and GSPO (Zheng et al., 2025).

**Evaluation Metrics.**   Lacking a faithful user simulator for end-to-end evaluation, we devise a suite of proxy metrics grounded in our hindsight framework. These metrics measure fine-grained alignment with expert strategy, serving as strong indicators of task success.

- **Strategic Questioning Quality (WA & WA-GH):** To measure *what to ask*, we report the average graded score (**WA**, for What-to-Ask) of generated questions on turns where continuing the dialogue is the correct action. This assesses if the agent targets the same critical information $I_t^*$ as the expert. We also report **WA-GH** (Good Hit rate), the proportion of these questions that achieve a perfect score, measuring the model's ability to generate excellent, precise questions. High scores on these metrics serve as a proxy for achieving high **Information Coverage**.
- **Dialogue Termination Accuracy (WS):** To measure *when to stop*, we report the accuracy (**WS**, for When-to-Stop) of the model's termination decision ('STOP') specifically on turns where the information-gathering goal has been met ($I_t^* = \emptyset$). A high WS score is a direct proxy for **Dialogue Efficiency** and the ability to avoid user fatigue.

Additionally, we report Dialogue Continuation Accuracy (when-to-continue, **WC**), overall Assessment Accuracy (**AA**) across all turns, Format Correctness (**FC**), and the final integrated Total Reward (**TR**). Detailed mathematical formulations for all metrics are provided in Appendix F.6.

## 4.2   MAIN RESULTS AND ANALYSIS

**`Learn-to-Ask` Excels at Policy Learning.**   We summarized our main results in Tab. 1. The primary finding is that our framework successfully teaches models both *what to ask* and *when to stop*. Compared to the base models, **Ours (GRPO)** shows dramatic gains. On the 7B model, the good-question hit rate (**WA-GH**) soars from 0.13 to 0.41 (**+215% rel.**), and termination accuracy (**WS**) jumps from 0.16 to 0.93. A similar trend holds for the 32B model, where **WA-GH** improves from 0.13 to 0.37 (**+185% rel.**) and **WS** from 0.52 to 0.88. This confirms that our hindsight-driven, decomposed reward structure is highly effective for learning a comprehensive dialogue policy.

**Qualitative Analysis.**   This quantitative effectiveness is mirrored in qualitative examples. As shown in Fig. 2, the SFT model asks an irrelevant question, as such a context may not be covered in the training data. In contrast, our model demonstrates strategic adaptation: it correctly identifies the information already provided and moves to an insightful follow-up. This highlights a shift from brittle mimicry to flexible, goal-oriented reasoning.

**Limits of Baselines and Nuances of Scale.**   The performance of our baselines underscores the difficulty of the task. **SFT** fails to generalize, sacrificing question quality (**WA** drops on both models) for rote memorization of stopping behavior. **DPO** collapses entirely on the 32B model, as its single binary preference signal is insufficient to guide the learning of our dual objectives. Interestingly, our own method shows slightly weaker results on the 32B model compared to the 7B one within this dataset. We attribute this to the limited data scale, which may not be sufficient to fully leverage the larger model's capacity. This is corroborated in our large-scale deployment (Sec. 5), where the 32B model's superiority becomes evident with ample data and more challenging business demands.

**Ablations and Further Analysis on Extensibility.**   Our ablation studies validate our design choices. Removing either the question reward (**w/o** $R_a$) or the stopping reward (**w/o** $R_s$) leads to a collapse in the corresponding skill, confirming the necessity of our dual-reward system. The multiplicative reward fusion also consistently provides a slight edge over simple summation, a benefit that is magnified in our complex production environment. For brevity, we defer more detailed analysis of alternative RL optimizers (e.g., CISPO, which outperforms other used RFT algorithms) to Appendix F.7, the model's performance on 9 public benchmarks with additional 14 metrics to Appendix H, and different hyperparameters and variations of SFT to Appendix F.8. In short, our method preserves general capabilities while being compatible with more advanced RFT algorithms.

**Tuning of $\beta$.** In reinforcement fine-tuning scenarios involving multiple reward signals, the strategy for reward integration critically influences the performance of trained models. In `Learn-to-Ask`, the tunable parameter $\beta$ plays a non-trivial role in the proposed multiplicative integration method. However, finding the optimal $\beta$ value is non-trivial. In production-level training, we find that the ideal value of $\beta$ depends on multiple factors—including the base models, the design of the reward models, etc., all of which may exert direct or indirect effects on the underlying distribution of reward signals. In our experiments, one may observe that increasing $\beta$ can simultaneously improve certain metrics while degrading others, reflecting a delicate trade-off. Consequently, we recommend selecting $\beta$ through empirical validation tailored to the specific task and data context.

**Data Quality.** Our framework assumes that the training data faithfully reflects the target interaction patterns — that is, all questions $a_t^*$ and assessments $I_t^*$ are accurate and reliable. To probe robustness against data corruption, we conduct a controlled experiment: we randomly select 20%, 40%, and 60% of the training samples and shuffle their ground-truth $a_t^*$ and $I_t^*$ entries, thereby constructing three increasingly noisy datasets. `Learn-to-Ask` demonstrates remarkable resilience: even with 60% corrupted samples, **WA** performance remains substantially above baseline. However, misaligned $a_t^*$ labels significantly degrade **WS** (When-to-Stop) performance, underscoring the importance of accurate action annotations for response generation.

Table 1: Main results on `Qwen2.5-7/32B-Instruct` models. Bold, underlined values indicate the best, second-best results among the baselines and our method, respectively.

| Model | | Qwen2.5-7B-Instruct | | | | | | | Qwen2.5-32B-Instruct | | | | | |
|---|---|---|---|---|---|---|---|---|---|---|---|---|---|---|
| Method | WA | WA-GH | WC | WS | AA | FC | TR | WA | WA-GH | WC | WS | AA | FC | TR |
| Base | 0.50 | 0.13 | **0.98** | 0.16 | 0.75 | 0.63 | 2.17 | 0.50 | 0.13 | 0.92 | 0.52 | 0.81 | 0.67 | 2.43 |
| SFT | 0.40 | 0.08 | 0.94 | 0.74 | 0.89 | 0.57 | 2.41 | 0.43 | 0.11 | **0.94** | 0.87 | **0.93** | 0.69 | 2.70 |
| DPO | 0.42 | 0.05 | 0.94 | 0.36 | 0.78 | 0.19 | 1.78 | 0.23 | 0.04 | 0.52 | 0.87 | 0.62 | 0.18 | 1.61 |
| **Ours** | **0.67** | **0.41** | 0.94 | **0.93** | **0.94** | **0.92** | **3.27** | **0.64** | **0.37** | 0.93 | **0.88** | 0.92 | **0.88** | **3.15** |
| *Ablation Studies* | | | | | | | | | | | | | | |
| w/o $R_s^*$ | 0.63 | 0.34 | 1.00 | 0.02 | 0.73 | 0.70 | 2.35 | 0.57 | 0.26 | 0.97 | 0.33 | 0.79 | 0.74 | 2.52 |
| w/o $R_a^*$ | 0.52 | 0.19 | 0.96 | 0.87 | 0.93 | 0.92 | 3.06 | 0.54 | 0.19 | 0.95 | 0.91 | 0.94 | 0.92 | 3.12 |
| Sum | 0.64 | 0.38 | 0.92 | 0.95 | 0.93 | 0.91 | 3.20 | 0.65 | 0.37 | 0.94 | 0.88 | 0.92 | 0.90 | 3.19 |
| *`Learn-to-Ask` with other RL algorithms* | | | | | | | | | | | | | | |
| GSPO | 0.61 | 0.31 | 0.93 | 0.94 | 0.93 | 0.91 | 3.16 | 0.62 | 0.32 | 0.95 | 0.86 | 0.93 | 0.89 | 3.12 |
| CISPO | 0.71 | 0.47 | 0.95 | 0.94 | 0.95 | 0.93 | 3.36 | 0.70 | 0.49 | 0.94 | 0.89 | 0.93 | 0.92 | 3.29 |
| *Swiping $\beta$* | | | | | | | | | | | | | | |
| $\beta = 1$ | 0.61 | 0.37 | 0.96 | 0.92 | 0.95 | 0.92 | 3.24 | 0.61 | 0.37 | 0.94 | 0.93 | 0.94 | 0.91 | 3.20 |
| $\beta = 2$ | 0.67 | 0.41 | 0.94 | 0.93 | 0.94 | 0.92 | 3.27 | 0.64 | 0.37 | 0.93 | 0.88 | 0.92 | 0.88 | 3.15 |
| $\beta = 4$ | 0.59 | 0.35 | 0.95 | 0.90 | 0.93 | 0.90 | 3.15 | 0.61 | 0.39 | 0.93 | 0.93 | 0.93 | 0.89 | 3.16 |
| *Data Quality* | | | | | | | | | | | | | | |
| 20% | 0.64 | 0.43 | 0.99 | 0.49 | 0.85 | 0.82 | 2.86 | 0.65 | 0.43 | 0.98 | 0.46 | 0.83 | 0.79 | 2.79 |
| 40% | 0.64 | 0.42 | 1.00 | 0.00 | 0.71 | 0.68 | 2.31 | 0.63 | 0.41 | 1.00 | 0.01 | 0.72 | 0.68 | 2.30 |
| 60% | 0.62 | 0.39 | 1.00 | 0.00 | 0.71 | 0.68 | 2.31 | 0.62 | 0.39 | 1.00 | 0.00 | 0.71 | 0.68 | 2.27 |

## 5 REAL-WORLD DEPLOYMENT AND IMPACT

**Deployment Contexts.** The ultimate validation of our framework is its ability to transition from offline logs to live, impactful applications. We successfully deployed a model trained with `Learn-to-Ask` in a large-scale online AI service with thousands of users daily (still growing), ``Medication AI Assistant'', whose goal is to proactively engage with the user to obtain a complete description of symptoms and recommend appropriate over-the-counter (OTC) medications.

**Model Scale in Production.** In our large-scale production environment, which involves a dataset over $100\times$ larger and covering $10\times$ more medical conditions than `RealMedConv`, the full capacity of larger models becomes essential. In this setting, the 32B model significantly outperforms the 7B model in both questioning quality and strategic accuracy, confirming that the saturation trend of

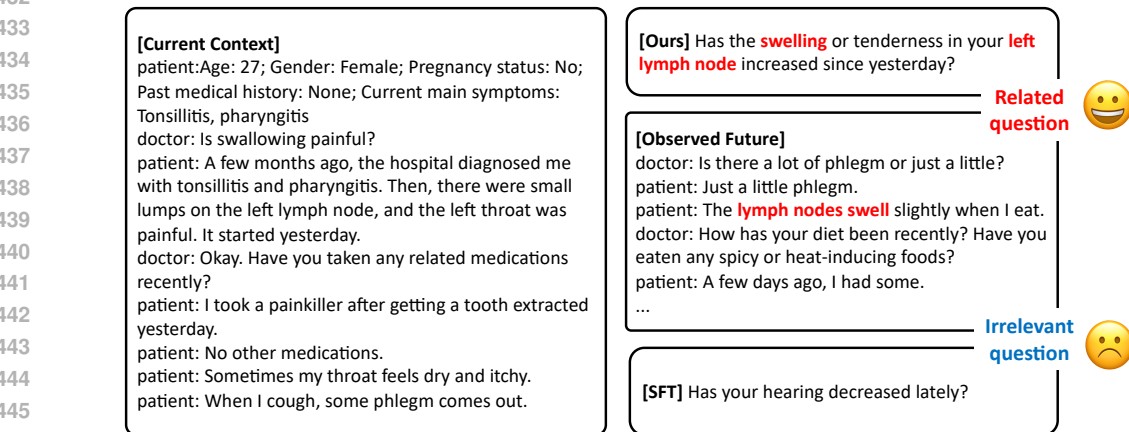

Figure 2: A case study comparing dialogues generated by SFT and `Learn-to-Ask` models.

performance boost observed on the small academic dataset does not apply to complex, larger-scale scenarios. We finally selected the 32B model for production deployment.

**The Role and Value of Auto-Prompt.** The Auto-Prompt pipeline was instrumental in our production system. In our offline experiments (see Appendix I for full results), calibrating the policy sampler prompt yielded relatively marginal gains (e.g., **TR** on 32B increased slightly from 3.145 to 3.166). This is likely because the academic task's simple prompt space can be effectively covered by manual tuning. However, in production, this automated approach becomes indispensable. Its true strength lies in the maintainability and continuous improvement it enables for the **extractor and grader prompts**. In the live system, we periodically identify ambiguous or low-performing online cases. These "margin examples" are then reviewed by human experts and added to the anchor sets. This allows us to re-calibrate our reward model and retrain the policy in a data-driven, semi-automated loop. This process ensures the agent adapts to evolving user behaviors and new business needs (e.g., incorporating new safety guidelines into the grader's logic) without costly and error-prone manual prompt engineering cycles. Auto-Prompt transforms a static training process into a dynamic, self-improving system.

**Online Performance and Validation of Proxy Metrics.** To rigorously evaluate the deployed model, we conducted a four-week live A/B test, routing a significant portion of user traffic to our model while a control group was served by the previous production model. The evaluation process was hybrid, involving both automated and human-led quality checks. Our model achieved **93% information completeness rate (ICR)** and an **88% good-question rate (GQR)**, [2] which are the online analogs to our offline WS and WA metrics. In addition to these strong internal scores, we measured the dialog-to-purchase conversion rate, a key business metric. Here, our model produced a lift ($\times 1.87$) compared to historical data from a parallel human-based service. These results provide powerful empirical evidence that our internal metrics are effective proxies for end-to-end task success and confirm the effectiveness of the proposed `Learn-to-Ask` framework.

## 6 DISCUSSION AND CONCLUSION

In this work, we introduced `Learn-to-Ask`, a general and simulator-free framework that bridges the "reality gap" in training proactive LLMs. By reframing the intractable long-horizon offline RL problem into a sequence of supervised tasks, our method learns a complete dialogue policy—including both what to ask and when to stop—directly from offline expert conversation logs. Our key insight is to leverage the *observed future* of each real trajectory to infer a dense and grounded reward signal, sidestepping the need for brittle user simulators.

---

[2]ICR is the ratio of conversations that covered sufficient information by the end, and GQR is the ratio of generated questions that are suitable for the context and aligned with human-experience, both are rated by qualified professionals.

Empirically, on a real-world medical dialogue dataset, `Learn-to-Ask` significantly outperformed strong baselines like SFT and DPO, demonstrating its superior ability to learn nuanced, strategic questioning. The framework's true value was validated by its successful deployment in a large-scale, commercial medical AI service, where our model achieved performance comparable to human experts and delivered tangible business impact. This provides powerful evidence that our offline proxy metrics translate directly to real-world task success.

**Generalization to Legal Domain.** To demonstrate the generalization capability of `Learn-to-Ask` beyond health care, we further evaluated our framework in the high-stakes legal domain. Due to the scarcity of publicly available, multi-turn dialogue datasets in this area, we leveraged the `CAIL` dataset [3] to construct a synthetic dialogue environment. We transformed real legal cases into conversational trajectories between a simulated lawyer and a suspect, where the lawyer proactively poses clarifying questions to infer the likely judgment. Despite the reliance on synthetic data, `Learn-to-Ask` consistently outperformed base models across all metrics, demonstrating significant gains in both conversational quality and assessment accuracy. These results provide further preliminary evidence of `Learn-to-Ask`'s cross-domain adaptability. Please refer to Appendix F.8 for detailed experimental setups and results.

**Theoretical Implications.** Beyond its practical utility, our work opens several new research avenues by connecting hindsight-based RFT to fundamental theories. Our framework can be seen as a stable, value-function-free offline RL algorithm, which raises a key question: *Can we formally characterize the sub-optimality gap of this hindsight-based policy compared to the true offline optimum?* From a causal perspective, we are heuristically learning an intervention policy. This invites future work on integrating do-calculus or counterfactual reasoning models to evolve from imitating optimal outcomes to predicting outcomes of *novel, unseen* interventions. Finally, our data-driven proxy for information gain and graph viewpoint suggests a new direction: *Could we learn to dynamically adjust the reward function itself to explore lines of inquiry not even present in the expert data, but which the theoretical information model deems valuable?* We hope these theoretical connections shed light on deeper analysis for the next generation of proactive agents. A detailed discussion is provided in Sec. D.

**Ethical Considerations.** Our framework is designed as an assistive tool to augment, not replace, human expertise. In high-stakes domains like healthcare and law, deployed models must operate under strict human-in-the-loop oversight to ensure safety, accountability, and compliance with professional standards. While `Learn-to-Ask` shows promising performance, outputs can still be inaccurate. Therefore, critical decisions should always be verified by qualified professionals, and deployment must include robust safeguards against misuse, such as access controls and continuous output auditing.

**From Imitation to Superhuman Intervention.** The most exciting frontier this work opens is the transition from expert imitation to superhuman AI agents. Our current model inherits human expert biases, such as a preference for conversational brevity (e.g., they tended to complete inquiries in a brisk 3-5 turns). Several directions evolved: (1) *Reward Shaping for Specific Goals:* Instead of merely rewarding coverage of the expert's information set $I_t^*$, future work can explore reward functions to enforce desired superhuman behaviors. For instance, we could add a penalty for any dialogue that concludes without explicitly asking a critical safety-related question (e.g., about allergies), even if the human expert omitted it. This allows for encoding organizational knowledge or safety protocols directly into the agent's policy. (2) *Exploration in Semantic Space:* A major challenge is to enable exploration without a live simulator. We can use a generator model to propose alternative, plausible information goals ($I_t'$) beyond the observed $I_t^*$. An advanced reward model, potentially trained on broader medical knowledge, could then score these hypothetical goals, allowing the agent to learn to pursue lines of inquiry that are valid but simply not represented in the limited offline dataset. (3) *Hybrid Human-AI Policy Learning:* The ultimate goal is not to replace human experts, but to augment them. Future systems could use our framework in an online loop. The AI can propose questions, and if a human expert overrules and asks something different, this action and its future outcome can be immediately incorporated to refine the AI's policy. This creates a symbiotic system where the AI continuously learns from and adapts to the evolving strategies of its human partners.

---

[3]https://github.com/china-ai-law-challenge/CAIL2018

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

**Ethics Statement** All authors have read and adhered to the ICLR 2026 Code of Ethics. Our research focuses on the algorithmic efficiency of reinforcement finetuning for Large Language Models and does not involve human subjects, animal experiments, or the processing of personally identifiable information. The datasets used in our experiments are publicly available and established benchmarks within the research community; all software, datasets, and frameworks utilized are governed by the permissive Apache-2.0 open-source license. Our method aims to make AI research more sustainable and accessible, and we do not foresee any direct negative societal impacts or ethical concerns arising from our proposed methodology. The authors declare no conflict of interest.

**Table of Contents**

## A    USAGE OF LARGE LANGUAGE MODELS

We employed LLMs solely for the purpose of grammar and typo checking in this manuscript, with Qwen3-235B-A22B and Gemini-2.5-pro. Their function was limited to tasks such as correcting grammatical errors, rephrasing sentences to enhance clarity and flow, and ensuring the consistent use of terminology. The LLMs had no role in the ideation of the research, the development of the proposed framework, the experimental design, or the analysis of results.

## B    DETAILED DISCUSSION ON RELATED WORKS

**Evolving LLMs as Proactive Agents.** Early dialogue systems explored proactive behaviors through rule-based or statistical methods, often in narrow domains (Deng et al., 2023a; Ling et al., 2025). The advent of LLMs shifted the focus towards leveraging their vast world knowledge. Initial efforts used prompting to elicit proactive behaviors like asking clarifying questions (Deng et al., 2023b; Zhao & Dou, 2024) or initiating topics (Liao et al., 2023). While straightforward, these methods lack the adaptability to learn complex, domain-specific strategies from data, a gap our training-based framework directly addresses.

**LLM Alignment for Single-Turn Attributes.** A popular fine-tuning paradigm focuses on improving single-turn response quality. This involves defining desirable attributes (e.g., relevance, clarity, safety) and training models on preference data, often synthetic, to align with these attributes (Zhou et al., 2022; Li et al., 2025b; Qian et al., 2023; Xu et al., 2025). These methods, including DPO and its variants, excel at local optimization. However, they are not designed to learn a long-horizon, stateful *policy*. Our work differs by framing the problem sequentially, learning not just *what* to ask but also the critical, policy-dependent decision of *when to stop*.

**LLM Alignment via Simulation and RL.** To tackle sequential decision-making, some works employ reinforcement learning in simulated environments (Xu et al., 2023; Wu et al., 2025). These approaches train an agent to interact with a user simulator to maximize a long-term reward. Their primary limitation is the simulator itself. Creating a realistic simulator for complex, open-ended domains like medical consultation is a monumental challenge. Another category of data simulation is to synthesize story-related reasoning tasks such as detective cases and situation puzzles by tree-based extension (Zhou et al., 2025). Policies trained in simulation often overfit to the simulator's quirks, leading to poor performance in the real world—the well-known "reality gap" (Hao et al., 2024). Our *Learn-to-Ask* framework is fundamentally simulator-free, learning directly from offline expert trajectories to ensure real-world applicability.

**Offline RL from Human Data.** Our work is philosophically aligned with offline reinforcement learning from human-involved data. Unlike standard offline RL, which assumes a fixed reward function, our key challenge is to *infer* the reward signal itself from expert behavior. Recent works have explored learning from trajectory-level preferences (Shi et al., 2024; Shani et al., 2024; Zhou et al., 2024). Our approach is distinct in its methodology: we decompose the long trajectory into single-turn decisions and infer fine-grained, turn-level rewards by using the *observed future* of the real conversation as a grounded source of truth. This allows for more precise and data-efficient policy learning.

**Connection to Hindsight and Goal-Conditioned Learning.** Our approach of using the observed future to define turn-level goals is philosophically related to Hindsight Experience Replay (HER) (Andrychowicz et al., 2017). HER relabels past experiences with goals achieved later in a trajectory to improve sample efficiency in sparse-reward RL. However, our work diverges in several critical aspects. First, we apply this concept to the complex, high-dimensional space of natural language dialogue, where goals are not simple state vectors but structured sets of semantic information ($I_t^*$) that must be dynamically extracted by an LLM. Second, standard HER focuses on reaching a goal state, whereas our framework learns a complete policy that includes an explicit, data-driven stopping condition ($s_t$), addressing the crucial question of *when* a goal is met. Thus, we view our contribution as a novel adaptation and significant extension of the hindsight learning paradigm to the domain of proactive LLM agents.

## C  Detailed Formulation on the Goal-Oriented Dialogue

### C.1  The Intuition From Information Graph Modeling

In Section 3.2, we introduced the conceptual model of a conversation as a flexible traversal of an implicit information graph. Here, we provide a more formal, albeit abstract, intuition for this model to further clarify the limitations of conventional fine-tuning methods.

Let a specific conversational goal (e.g., diagnosing a particular condition) be associated with an underlying **information graph** $\mathcal{G} = (\mathcal{V}, \mathcal{E})$.

- The set of **vertices** $\mathcal{V}$ represents all potentially critical pieces of information (or "information nodes") needed to satisfy the goal. For example, in a cold diagnosis, vertices might include 'v_fever_status', 'v_cough_type', 'v_symptom_duration', etc. A special vertex, $v_{\text{start}}$, represents the initial user query.

- The set of **directed edges** $\mathcal{E}$ represents dependencies between information nodes. An edge from $v_i$ to $v_j$ implies that question $q_j$ (which aims to uncover information $v_j$) is a natural follow-up to question $q_i$. For instance, after confirming the presence of a cough ('v_cough_present'), an edge might lead to inquiring about its type ('v_cough_type'). Many nodes may be directly reachable from $v_{start}$, representing independent lines of inquiry.

An expert's conversation trajectory, $\tau$, can be viewed as a specific path or walk through this graph, starting from $v_{start}$. The expert's policy aims to select a sequence of questions that efficiently covers a **sufficient subgraph** of $\mathcal{G}$—a set of nodes whose information, taken together, is enough to make a final decision (e.g., recommend a medication).

To better understand our model, we present an illustrative example as shown in Fig. 3. Given a conversation trajectory on the left, with our model, we can define $v_{\text{start}} = $ "male, 35 years old, having a cold for 2 days", nodes $v_A = $ "Do you have fever?", $v_B = $ "What is your temperature?", $v_C = $ "Do you cough?", and edge $e_{AB} = $ "yes". Clearly, in this example, trajectory $\tau_1 = (S \rightarrow A \rightarrow B \rightarrow C)$ can be formulated as a graph $\mathcal{G}$ as shown in the figure on the right. Obviously, given a trajectory $\tau_2 = (S \rightarrow C \rightarrow A \rightarrow B)$, we could derive the identical graph. Therefore, start from $v_{\text{start}}$, the space for the next question in this case is a set $\{v_A, v_B\}$.

**How this model exposes the weakness of SFT and DPO:**  This graph-theoretic perspective crystallizes why myopic, single-step optimization methods are insufficient:

- **SFT learns edges, not coverage.** SFT trains the model to predict the next node in one specific, observed path. In the example in Fig. 3, if the training samples are built upon $\tau_1$, the learned policy would always go from node S to A. It has no mechanism to understand that going from S to B might be an equally valid or even better choice in a different context. It lacks the notion of "set coverage" and is confined to memorizing paths.

- **DPO struggles with path ambiguity.** Using the same example, we know both $\tau_1$ and $\tau_2$ are valid trajectories. If we create DPO data from $\tau_1$, we might generate a preference pair where 'A' is chosen over 'C'. If we do the same for $\tau_2$, we might generate a pair where 'C' is chosen over 'A'. When trained on a large dataset containing both types of trajectories, the DPO objective receives conflicting preference signals for the same state 'A', making it difficult to learn a coherent, globally optimal policy. The preference is path-dependent, but DPO treats it as a local, path-independent signal.

In contrast, our **Learn-to-Ask** framework is designed to address this. By using the observed future to define a target information set $I_t^*$, our method effectively estimates the "remaining nodes to be covered" from the current state. This provides a global, coverage-based learning signal that is robust to the specific path taken, thereby overcoming the myopia of SFT and the ambiguity of DPO.

### C.2  Challenges of Offline RL in Goal-Oriented Dialogue

In Section 3.2, we noted that applying offline RL to dialogue faces two major challenges. Here, we provide a more detailed exposition.

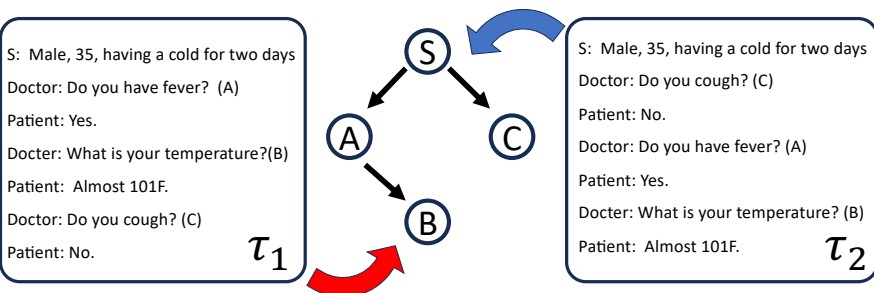

Figure 3: An illustrative example of the conceptual graph model.

**1. The Simulator Gap and The Reality Gap**  Online RL algorithms like PPO improve a policy by actively interacting with an environment to collect new data. In dialogue, this would require a user simulator. However, building a high-fidelity user simulator that can realistically respond to any question in an open-ended, expert domain (like medicine or law) is an unsolved and monumental task (Wu et al., 2025). A simplistic simulator would lead to the agent over-exploiting its flaws. The resulting policy, when deployed in the real world, would likely fail due to the distribution shift between the synthetic and real user behavior—a phenomenon known as the "reality gap" (**?**). Our simulator-free approach completely bypasses this problem.

**2. Instability of Offline Value Estimation**  Offline RL algorithms must learn from a fixed, static dataset. Many prominent methods, such as those based on Q-learning (e.g., CQL (Kumar et al., 2020)), aim to learn a state-action value function $Q(s, a)$. In the context of dialogue, the state space (all possible conversation histories) and action space (all possible questions) are effectively infinite and compositional. This poses a severe problem for value-based methods:

- **Extrapolation Error:** The Q-function must be queried for actions that may not be present in the offline dataset (out-of-distribution actions). Neural networks are notoriously bad at this, often producing arbitrarily high and erroneous Q-values for unseen actions (Fujimoto et al., 2019).
- **Divergence:** A policy trying to maximize these overestimated Q-values will choose poor actions, leading to a "bootstrapping error" where the Bellman update further corrupts the value function. This can cause the entire training process to diverge.

While methods like CQL add conservative penalties to mitigate this, they are often complex to tune and can be overly pessimistic. Our approach of reframing the problem as supervised learning on hindsight-based objectives avoids the need to estimate a long-horizon, unstable value function altogether, leading to a much more stable and direct learning process.

## D    THEORETICAL PERSPECTIVES ON LEARN-TO-ASK

Our empirical success motivates a deeper theoretical examination of why Learn-to-Ask is effective. Here, we analyze our framework from three perspectives: offline reinforcement learning, causal inference, and information theory. These discussions frame our work within established theoretical paradigms and highlight its novel contributions.

### D.1    AS A VALUE-FUNCTION-FREE OFFLINE RL PARADIGM

The predominant challenge in offline reinforcement learning is *extrapolation error*, where a learned value function (e.g., Q-function) produces arbitrarily high, erroneous values for out-of-distribution (OOD) actions not present in the static dataset (Fujimoto et al., 2019; Kumar et al., 2020). This leads to policy divergence, as the agent learns to exploit its own value function's flaws. State-of-the-art offline RL algorithms combat this by introducing explicit pessimism, either by constraining the pol-

icy to stay close to the data-generating behavior policy (policy-based constraints) or by regularizing the value function to assign low values to OOD actions (value-based constraints).

`Learn-to-Ask` sidesteps this central problem entirely by being a **model-free and value-function-free** algorithm. It never learns an explicit state-action value function $Q(C_t, a_t)$. Consequently, it is immune to extrapolation error by design. Instead of answering the counterfactual question, "What would be the long-term value if I took action $a_t$?", our framework answers a more direct, hindsight-grounded question: "Given that a successful expert ultimately achieved goal set $I_t^*$ from this state, what action $a_t$ aligns with this revealed objective?"

This reframing comes with an implicit but powerful assumption: *the future sequence of actions in an expert trajectory constitutes a near-optimal plan from the current state.* Our hindsight inference process effectively treats the outcome of this plan (the collected information $I_t^*$) as a direct supervisory signal. This can be viewed as a practical and highly scalable simplification of Inverse Reinforcement Learning (IRL). Rather than undertaking the full, often intractable, task of learning a general reward function from expert demonstrations, we assume a specific, task-oriented reward structure—maximizing coverage of the 'to-be-collected' information set—and directly use it for policy optimization. This approach trades generality for stability and scalability, providing a robust blueprint for offline policy learning in high-dimensional, structured action spaces like natural language.

**Future Potentials.** This value-function-free perspective opens several research questions. First, can we derive a theoretical bound on the sub-optimality of the policy learned via `Learn-to-Ask` with respect to the true optimal offline policy? This would likely depend on the "quality" or "coverage" of the expert data. Second, while our method avoids value overestimation, it is inherently limited to the outcomes observed in the data. A hybrid approach could be promising: using our stable, hindsight-driven policy as a base and then performing a cautious, value-based policy improvement step on top of it to discover slightly out-of-distribution but superior actions.

## D.2 As a Heuristic for Causal Intervention

The task of proactive questioning can be framed as a problem of sequential causal inference. At each turn $t$, the agent seeks to choose an action (a question, or *intervention*) $a_t$ that maximizes a desired future outcome (e.g., task success, information completeness). In the potential outcomes framework (Rubin, 1974), for each possible action $a_j \in \mathcal{A}$, there exists a potential outcome $Y(a_j)$ representing the state of the world had we intervened with $a_j$. The agent's goal is to select $a^* = \arg\max_{a_j} \text{Utility}(Y(a_j))$. The fundamental problem of causal inference is that we can only ever observe one of these potential outcomes for any given instance—the one corresponding to the action actually taken.

Standard supervised methods like SFT operate in a purely observational regime. They learn a policy $\pi(a_t|C_{t-1})$ that mimics the expert's chosen action $a_e$, but they have no model of the causal link between the action $a_e$ and its outcome $Y(a_e)$. They are learning correlation, not causation.

`Learn-to-Ask` offers a powerful heuristic to approximate causal reasoning. It operates on the core assumption that *the expert's trajectory represents a sequence of near-optimal interventions.* By extracting the future information set $I_t^*$, our method essentially reconstructs the outcome $Y(a_e)$ that the expert's intervention $a_e$ was designed to achieve. The policy is then trained not just to mimic $a_e$, but to generate actions that are effective at achieving the *goal* $Y(a_e)$. This encourages the model to learn a rudimentary understanding of the action-outcome relationship. While it does not allow for true counterfactual reasoning (i.e., estimating $Y(a_k)$ for an unobserved action $a_k$), it moves beyond simple behavioral cloning towards goal-conditioned behavioral learning, which is a step closer to learning a causal policy from offline observational data.

**Future Potentials.** The connection to causal heuristics suggests a path toward more powerful reasoning. A significant future direction is to move from our current heuristic to a more formal causal model. For instance, could we use the offline data to build a structural causal model (SCM) of the dialogue, where questions are interventions and user responses are outcomes? Such a model, even if approximate, could enable true counterfactual queries, allowing the agent to ask "What would the user have said if I had asked about 'headaches' instead of 'fever'?" Answering such questions

would unlock the ability to plan and act in truly novel situations not covered by the expert data, representing a leap from imitation to genuine strategic reasoning.

### D.3 AS A DATA-DRIVEN PROXY FOR INFORMATION GAIN

From an information-theoretic perspective, an ideal proactive agent should, at each turn, select the question that maximizes the **expected information gain** about the user's underlying state (e.g., their true medical condition). This is equivalent to maximizing the mutual information between the question-answer pair and the latent user state. However, in open-ended domains, defining the latent state space and the associated probability distributions is intractable, making direct computation of information gain impossible.

`Learn-to-Ask` provides a pragmatic, data-driven proxy for this principle. It relies on the hypothesis that *human experts, through years of experience, develop an intuitive policy that is highly effective at maximizing information gain.* Their line of questioning is not random; it is structured to efficiently reduce uncertainty.

Our framework operationalizes this hypothesis. The hindsight inference of the target information set, $I_t^* = \text{Extract}(...)$, can be interpreted as a procedure to *decode the expert's implicit, high-information-gain targets.* Instead of computing an abstract information-theoretic quantity, we directly identify what a real expert deemed was the most critical information to acquire next. The subsequent policy learning then trains the agent to align its actions with these empirically-grounded, high-value information targets. In essence, `Learn-to-Ask` substitutes the analytically intractable problem of maximizing a theoretical information metric with the tractable, data-driven problem of aligning with an expert's revealed information-seeking intent.

**Future Potentials.** Viewing our method as a proxy for information gain invites research on closing the gap with the true theoretical principle. One avenue is to develop a "semantic uncertainty" model. Instead of a full probabilistic model of the user state, an LLM could be trained to estimate its own uncertainty over a set of predefined clinical entities. The policy could then be rewarded for asking questions that are predicted to reduce this uncertainty metric the most. A more ambitious goal would be to integrate our hindsight-based reward with an uncertainty-based reward term, creating a policy that both grounds itself in proven expert strategies and actively seeks to reduce its own knowledge gaps.

### D.4 FURTHER DISCUSSION ON THE GRAPH-THEORETIC MODEL

As introduced in Section C.1, we can conceptualize goal-oriented dialogue as a traversal of an implicit information graph $\mathcal{G} = (\mathcal{V}, \mathcal{E})$. `Learn-to-Ask` fundamentally alters the learning objective compared to myopic methods.

**SFT and DPO Learn Edge Preferences:** Both Supervised Fine-Tuning (SFT) and Direct Preference Optimization (DPO) operate at the level of edge traversal. SFT learns a deterministic policy to traverse a specific edge (e.g., $A \rightarrow B$) if it appeared frequently in the training data. DPO learns a preference for one edge over another from a given node (e.g., preferring $A \rightarrow B$ over $A \rightarrow C$). Both are local and memory-based, lacking a concept of the global goal. They are prone to getting "stuck" if the conversation deviates from a memorized path, and they struggle to synthesize strategies from diverse expert trajectories that may have equally valid but different paths.

**`Learn-to-Ask` Learns a Subgraph Coverage Policy:** Our framework operates at a higher level of abstraction. At any node $v_t$ (representing the information in context $C_{t-1}$), the hindsight inference mechanism identifies the set of remaining critical nodes $\{v_i, v_j, ...\} = I_t^*$ that the expert eventually covered to complete a sufficient subgraph. The policy is then rewarded for any action $a_t$ that leads to the discovery of any node in this target set.

This has two profound advantages:

1. **Robustness to Path Variation:** It correctly learns that from node $A$, both edges $A \rightarrow B$ and $A \rightarrow C$ are valuable if both $B$ and $C$ are part of the required information subgraph. This allows the model to learn a more flexible and robust policy that generalizes across

the diverse strategies present in the expert data, rather than overfitting to the single most frequent path.

2. **Principled Stopping Condition:** The "when to stop" decision emerges naturally from this model. The agent learns to stop when the inferred target set $I_t^*$ is empty, which corresponds to the state where the sufficient information subgraph has been fully covered. This provides a goal-grounded, non-arbitrary mechanism for dialogue termination, a component critically absent in myopic, single-turn optimization methods.

In summary, `Learn-to-Ask` shifts the learning paradigm from "mimicking the next step" to "understanding the remaining goal," enabling it to learn a true, stateful policy directly from offline logs.

**Future Potentials.** The graph model itself presents opportunities for future work. Currently, the information graph $\mathcal{G}$ is implicit. An exciting research direction would be to learn this graph structure explicitly from data. By analyzing thousands of expert trajectories, one could potentially mine the latent dependency structure between information nodes (e.g., questions about 'cough type' often follow questions about 'fever'). If this latent graph could be constructed, it would serve as a powerful prior for policy learning. A new agent could be trained to traverse this graph efficiently, or even identify "holes" in the graph representing un-asked but potentially valuable questions, thus enabling a form of structured exploration.

# E   IMPLEMENTATION DETAILS FOR AUTO-PROMPT CALIBRATION

As illustrated in Algorithm 1, our pipeline is an iterative search process over the space of prompts, which is implemented based on Data-Juicer Sandbox (Chen et al.; 2025b). It operates on three parallel tracks for the Info-Extractor, the Reward Grader and the Policy Rollout, using a shared methodology but distinct objectives and calibration data.

---

**Algorithm 1** Automated Prompt Optimization

---

1: **Input:** Initial prompts $P_{\text{seed}}^0$, calibration sets $\mathcal{D}_{\text{calib}}$, human-verified anchor sets $\mathcal{D}_{\text{anchor}}$, number of iterations $K$, prompt type $T \in \{\text{EXTRACT}, \text{GRADER}, \text{ROLLOUT}\}$.
2: **Initialize:** Best prompts $P_{\text{best}} \leftarrow P_{\text{seed}}^0$.
3: **for** $k = 1, \dots, K$ **do**
4:     Generate candidate prompts $\mathcal{P}_{\text{cand}}$ from $P_{\text{best}}$.
5:     Execute type-specific pipelines for each candidate: $O_j = \texttt{Pipeline}(P_j, \mathcal{D}_{\text{calib}}, T)$
6:     Compute consistency score against labels from $\mathcal{D}_{\text{anchor}}$: $S_j = \texttt{Score}(O_j, \mathcal{D}_{\text{anchor}}, T)$
7:     Update $P_{\text{best}} \leftarrow \arg\max_{P_j} S_j$.                            ▷ Maximizing score
8: **end for**
9: **Output:** Calibrated prompts $P_{best}$.

---

The pipeline consists of four key steps, executed iteratively:

1. **Candidate Generation:** Starting with a seed prompt (for either the extractor or grader), a generator LLM proposes variations. These variations are created through semantic paraphrasing (e.g., "Rephrase this instruction to be more explicit about X") and rule-based mutations (e.g., adding or removing few-shot examples), exploring a diverse set of instructions.

2. **Type-specific Pipeline Execution on Calibration Set ($\mathcal{D}_{\textbf{calib}}$):** Each candidate prompt is used to execute a type-specific pipeline on a calibration dataset. This set, $\mathcal{D}_{\text{calib}}$, is designed to be flexible and can be tailored to specific business scenarios or challenging edge cases, ensuring the resulting prompts are robust for varied real-world situations. For different prompt types, different pipeline functions $\texttt{Pipeline}(P_j, \mathcal{D}_{\text{calib}}, T)$ are executed:

   - For the **Info-Extractor** ($T = \text{EXTRACT}$), the information set extraction pipeline is conducted on the calibration dataset $\mathcal{D}_{\text{calib}}$ with the candidate info-extractor prompt $P_j$. It returns the extracted information set as $O_j$.
   - For the **Reward Grader** ($T = \text{GRADER}$), the grader model with candidate grader prompt $P_j$ returns the rewards $O_j$ of the calibration dataset $\mathcal{D}_{\text{calib}}$ that contains prepared rollouts.

- For the **Policy Rollout** ($T = $ ROLLOUT), the policy model generates rollouts with the candidate rollout prompt $P_j$ on the calibration dataset $\mathcal{D}_{\text{calib}}$ and then the fixed grader computes the rewards $O_j$ on these policy rollouts.

3. **Consistency Scoring with Human Anchors ($\mathcal{D}_{\text{anchor}}$):** The quality of each candidate prompt is measured by its consistency with a small, high-quality, human-verified anchor set. Instead of requiring expensive, large-scale labeling, we use targeted human verification on a handful of ambiguous "margin examples." Different scoring methods $\text{Score}(O_j, \mathcal{D}_{\text{anchor}}, T)$ are used for different prompt types:

   - For the **Info-Extractor** ($T = $ EXTRACT), the consistency $S_j$ is measured by **accuracy** (e.g., F1-score or exact match) between extracted information sets $O_j$ and human-annotated information sets $\mathcal{D}_{\text{anchor}}$. The goal is to find the prompt that best reproduces the expert's information extraction.

   - For the **Reward Grader** ($T = $ GRADER), which outputs a continuous score $S_j$, consistency is measured by negative **Mean Squared Error (MSE)** between the grader outputs $O_j$ and human-assigned graded scores (e.g., 0.0, 0.5, 1.0). The goal is to find the prompt whose scoring logic most closely mimics a human evaluator's nuanced judgment.

   - For the **Policy Rollout** ($T = $ ROLLOUT), once the grader is settled, human anchors are not necessary. The goal is just to find the rollout prompt that generates the policy rollouts with the highest reward $O_j$ from the reward grader.

4. **Selection and Iteration:** The candidate prompt that demonstrates the highest consistency (accuracy for the extractor, negative MSE for the grader, grades for the rollout) is selected as the new best prompt for the next iteration. This entire loop can be run automatically until performance on a held-out validation set converges.

# F  EXPERIMENTAL DETAILS

## F.1  DATASET DETAILS AND PRE-TRAINING PREPARATION

The RealMedConv dataset is built from anonymized logs of real-world interactions between licensed pharmacists and users seeking over-the-counter medication advice. Each session has a clear goal: gather sufficient symptom information to make a safe and appropriate recommendation. The dialogues are typically 3-5 turns long, reflecting the efficient, goal-directed nature of expert interactions.

To prepare the training dataset for each experiment, for each full dialogue trajectory $\tau = (u_0, a_1, u_1, \ldots, u_{T-1})$, we first split the trajectory into the current context $C_{t-1} = (u_0, \ldots, u_{t-1})$ and the observed future $C_t^c = (a_t, u_t, \ldots, u_{T-1})$ at each $t \in [0, T-1]$. Next, for each experiment setting, we further process the segments as follows.

- **RL:** We then apply our hindsight pipeline (Section 3.4) to this pair to generate the ground-truth objective tuple $(I_t^*, s_t^*)$. This results in a training sample

$$\langle \text{input} = C_{t-1}, \text{reward reference} = (I_t^*, s_t^*) \rangle.$$

  Note that for the ablation study *without* $R_s$, we omit all samples with ground truth STOP, as there is no valid definition of reward $R_a$ for such samples. Similarly, all samples with ground truth $s^* = $ CONTINUE and $I^* = \emptyset$ are omitted as there is no valid $R_a$ for such cases.

- **SFT:** We take the immediate next assistant utterance as the expected response, and obtains sample

$$\langle \text{input} = C_{t-1}, \text{response} = a_t \rangle.$$

- **DPO:** We take the immediate next assistant utterance $a_t$ as the 'chosen', while using LLM to generate an utterance that is irrelevant to any content in the trajectory as 'rejected'. This results in a sample

$$\langle \text{input} = C_{t-1}, \text{chosen} = u_t, \text{rejected} = \text{some irrelevant utterance} \rangle.$$

### F.2 DETAILS FOR THE INFORMATION EXTRACTOR

The design of the information extractor is context-dependent. For example, in a diagnostic context, the task is to extract the facts that appear in the conversation to compose a complete symptom description, which can be done by utilizing powerful LLMs with appropriate prompts, which can be tuned by our `Auto-Prompt` to align with human expectations.

However, as mentioned in Section 3.4, to prevent the learned model from committing a reward hacking such as keeping asking overly generic but frequently occurring questions (e.g., pregnancy status), we need to avoid collecting context-independent or overly generic information in the resulting $I_t^*$. There are several practical solutions. For example, one may randomly pick a small number of samples (e.g., a few hundred), then compute the appearing frequency of each extracted information point. Any information point appearing over a certain threshold (e.g., 80%) is flagged as 'generic'. Alternatively, objective human observation across a number of samples can identify patterns fairly easily. For example, within reading a few hundred samples, one could easily find out that physicians often ask about allergies, used medication, past illness, or pregnancy status before they make a medication decision, since these questions are part of the standard procedure.

### F.3 DETAILS FOR EXTRA TERM $\Omega$

As introduced in Section 3.6, the term $\Omega$ could either be defined as a reward or a penalty. In this work, we chose to make it a penalty that controls the format of the output, which is defined as follows:

$$\Omega(a_t, s_t = \texttt{CONTINUE}) = \begin{cases} 1 & \text{if } R_s = 1, \text{ and } a_t \text{ contain exactly one question,} \\ 0.5 & \text{if } R_s = 1, \text{ and } a_t \text{ contain exactly two questions,} \\ 0, & \text{otherwise.} \end{cases}$$

And

$$\Omega(a_t, s_t = \texttt{STOP}) = \begin{cases} 1 & \text{if } R_s = 1 \text{ and } a_t = \langle STOP \rangle, \\ 0, & \text{otherwise.} \end{cases}$$

Here, some conditions can be evaluated by LLMs together with $R_a$, example prompts are given in Appendix G.

It is worth noting that $\Omega$ plays a crucial role in the training to regulate the output format. Here, we require generating exactly one question to avoid the "shotgun effect" (generate multiple questions to increase the chance of hitting valid information points in $I_t^*$ and getting a reward).

### F.4 IMPLEMENTATION DETAILS

All experiments were conducted on a cluster of up to 32 NVIDIA H20 GPUs. We utilized the `Trinity-RFT` framework (Pan et al., 2025), a highly customizable RFT training library, to implement our entire workflow, including policy sampling, reward grading, and optimization.

To ensure fair comparison, all primary hyperparameters (e.g., learning rate $= 5e^{-7}$, batch size $= 64$, number of training epochs $= 4$) were kept consistent across all methods and models. For group RL algorithms (i.e., GRPO, CISPO, GSPO), we take 5 repeats for each sample. Full parameter settings can be found in the configuration files in the released source code.

The policy-sampler prompt, info-extractor prompt, and reward-grader prompt were all calibrated using our Auto-Prompt pipeline (Section 3.5) before the main training runs.

### F.5 BASELINE AND ABLATION IMPLEMENTATION DETAILS

- **SFT and DPO:** These baselines use the prompt shown in Appendix G and datasets prepared as introduced in Appendix F.1.
- **Ablation (w/o $R_s$):** In this setting, the model was only trained on dialogue turns where the ground-truth action was `CONTINUE` as introduced in Appendix F.1. The system prompt for the policy was modified to only instruct question generation, removing any mention of the stopping condition. The reward was simplified to $R(a_t, s_t) = \beta \cdot R_a(a_t; I_t^*) + \Omega(a_t, s_t)$.

- **Ablation (w/o $R_a$):** The model was trained on the full dataset, but the reward function ignored the question quality, becoming $R(a_t, s_t) = R_s(s_t; s_t^*) + \Omega(a_t, s_t)$.

- **Ablation (Sum):** The reward function was changed to an additive form: $R(a_t, s_t) = R_s(s_t; s_t^*) + \beta \cdot R_a(a_t; I_t^*) + \Omega(a_t, s_t)$.

### F.6 METRIC DEFINITIONS

We calculate metrics aligning with our reward structure and measure the model's fine-grained capabilities.

- *What-to-Ask (WA):* This metric is the average $R_a^*$ score on samples whose ground truth $s^* = \texttt{CONTINUE}$, and the policy also correctly chose to continue the questioning. We also provide a variation *WA-GH (Good Hit)*, which is the proportion of generated results that achieve a full score, defined as

$$\text{WA-GH} = \frac{\text{total \# of correct } \texttt{CONTINUE} \text{ samples with } R_a^* = 1}{\text{total \# of correct } \texttt{CONTINUE} \text{ samples}}.$$

- *When-to-Continue (WC):* This metric is the average $R_s^*$ score on samples whose truth $s^* = \texttt{CONTINUE}$. It is worth noting that this metric is somehow misleading, as high **WC** may imply the policy is weak in making termination assessment – it only trivially chooses to continue the conversation. Nevertheless, we still keep this in the metrics for completeness.

- *When-to-Stop (WS):* In contrast to **WC**, this metric is the average $R_s^*$ score on samples whose truth $s^* = \texttt{STOP}$, and it particularly focuses on the capability of correctly terminating the questioning process.

- Other metrics including *Assessment-Accuracy* (AA), which is the average assessment score $R_s^*$; *FormatCorrectness* (FC), which is the average format score $P$; and *TotalReward* (TR), which is the average overall reward score integrated by Eq. 5, across all samples.

### F.7 LEARN-TO-ASK WITH OTHER RL ALGORITHMS.

Our experiments take GRPO (Shao et al. (2024)) as the prime optimization algorithm. We also report the evaluation of our method on some of the RL algorithms new to the literature, which are designed for better efficiency in training, for example, GSPO (Zheng et al. (2025)) and CISPO (Chen et al. (2025a)). As shown in Fig. 4, the algorithms display different training efficiency reflected by the reward growth rates. CISPO, an algorithm that clips importance sampling weights rather than token updates, is relatively faster than GRPO (ours). The evaluated results in Tab. 1 display the same pattern, within 4 epochs (385 steps) of training, CISPO obtained the best performance in learning what-to-ask, while maintaining performance similar to GRPO in learning when-to-stop. Nevertheless, there is still plenty of room for improving the overall performance by developing more efficient RL algorithms, and we would leave that for future work.

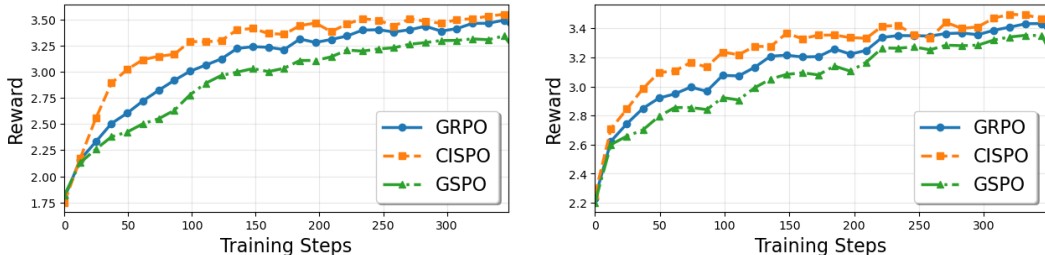

Figure 4: The reward growing curves of RL algorithms in training 7B (left) and 32B (right) models.

### F.8 ADDITIONAL EXPERIMENTAL RESULTS

In this section, we present additional experimental results in Table 2, regarding the following perspectives:

- **Supervised Fine-Tuning (SFT)**. In the main experiments, we report results at epoch $= 4$ in Table 1, which serves as the default training duration across all settings. To better assess the efficacy of SFT on this task, we further extend training up to epoch $= 8$. Our observations indicate that both the 7B and 32B models exhibit rapid performance gains during the initial epochs, followed by a pronounced slowdown in improvement. Notably, the **WA** (What-to-Ask) score even slightly declines beyond epoch 4 — a clear sign of overfitting, a well-known limitation of SFT when trained for overly many epochs.

- **Generalization to Other Domains**. While our primary evaluation focuses on medical conversations, readers may be interested in the applicability of `Learn-to-Ask` to other domains. While we have multiple ongoing in-house evaluations of our framework on domains other than medical conversation, due to confidentiality reasons, we cannot release more details regarding those attempts. Publicly available, high-quality, multi-turn conversational datasets in specialized domains, such as technical support, legal consulting, and other high-stakes domains, remain rare in the open-source domain —largely due to privacy and legal concerns. As a pragmatic alternative, we leverage the CAIL (Challenge of AI in Law) [4] dataset, which contains over 150K legal cases and associated judgments. We randomly sampled 1.8K cases, use a powerful LLM to convert each case into a synthetic dialogue between a suspect and a lawyer, where the lawyer iteratively asks clarifying questions to infer the likely sentence. Despite potential imperfections in these synthesized trajectories, Table 2 shows that `Learn-to-Ask` consistently outperforms base models across all metrics: both conversational quality (**WA** and **WA-GH**) and assessment accuracy (**WS**) see significant gains. This provides strong preliminary evidence of `Learn-to-Ask` 's cross-domain adaptability beyond the medical setting.

Table 2: Additional results on `Qwen2.5-7/32B-Instruct` models.

| Model | | Qwen2.5-7B-Instruct | | | | | | Qwen2.5-32B-Instruct | | | | | |
|---|---|---|---|---|---|---|---|---|---|---|---|---|---|
| Method | WA | WA-GH | WC | WS | AA | FC | TR | WA | WA-GH | WC | WS | AA | FC | TR |
| Base | 0.50 | 0.13 | 0.98 | 0.16 | 0.75 | 0.63 | 2.17 | 0.50 | 0.13 | 0.92 | 0.52 | 0.81 | 0.67 | 2.43 |
| **Ours** | **0.67** | **0.41** | **0.94** | **0.93** | **0.94** | **0.92** | **3.27** | **0.64** | **0.37** | **0.93** | **0.88** | **0.92** | **0.88** | **3.15** |
| | | SFT | | | | | | | | | | | | |
| epoch=2 | 0.40 | 0.08 | 0.93 | 0.62 | 0.85 | 0.53 | 2.25 | 0.45 | 0.11 | 0.89 | 0.82 | 0.87 | 0.69 | 2.58 |
| epoch=4 | 0.40 | 0.08 | 0.94 | 0.74 | 0.89 | 0.57 | 2.41 | 0.43 | 0.11 | 0.94 | 0.84 | 0.92 | 0.69 | 2.65 |
| epoch=8 | 0.37 | 0.08 | 0.94 | 0.79 | 0.90 | 0.58 | 2.44 | 0.44 | 0.16 | 0.95 | 0.89 | 0.94 | 0.76 | 2.80 |
| | | Learn-to-Ask in legal domain | | | | | | | | | | | | |
| Base | 0.39 | 0.08 | 0.98 | 0.31 | 0.83 | 0.60 | 2.17 | 0.43 | 0.09 | 0.97 | 0.41 | 0.85 | 0.71 | 2.38 |
| epoch=2 | 0.54 | 0.24 | 0.96 | 0.74 | 0.91 | 0.82 | 2.86 | 0.52 | 0.24 | 0.93 | 0.84 | 0.91 | 0.84 | 2.89 |
| epoch=4 | 0.62 | 0.39 | 0.95 | 0.84 | 0.93 | 0.87 | 3.09 | 0.61 | 0.36 | 0.90 | 0.95 | 0.91 | 0.87 | 3.06 |

# G USED PROMPTS

We present the specific seed prompt used in the extractor for target information set below.

```
[System] You are an expert information analyst. Your task is to identify the
new, goal-relevant information a professional gathered in a conversation.

[Goal] The user wants to find medication for a cold with a cough.

[Current Context]
User: "I have a cold and a bad cough."
Assistant: "Okay, I understand. To help you better, I need more details."

[Future Conversation]
Assistant: "Do you have a fever?"
User: "No, no fever."
Assistant: "Is your cough productive, meaning are you coughing up phlegm?"
User: "Yes, and it's yellow."
```

---

[4]https://github.com/china-ai-law-challenge/CAIL2018

```
[Instruction] Based on the [Future Conversation], list the critical new pieces
of medical information the assistant elicited from the user, which were not in
the [Current Context]. Output as a structured list.

[Expected Output]
- Information on fever (absent)
- Type of cough (productive)
- Color of phlegm (yellow)
```

The prompt for response generation in general:

```
[System] You are a medical assistant.
Your task is to understand the ongoing conversation and
continue the medical inquiry in English.

[Guidelines]
- Each response must contain exactly one clear and concise medical question
  with 2 to 3 answer choices.
- Do not repeat any previous question.
- Your response must be a single sentence.
- If enough information has been gathered to make a medication suggestion,
  output only: <stop />
```

The prompt for response generation in the ablation studies: *without $R_a^*$, SFT and DPO*:

```
[Task] You are a medical assistant.
Your task is to understand the ongoing conversation
and continue the medical inquiry in English.

[Guidelines]
- If enough information has been gathered to make a medication suggestion,
  output only: <stop />
```

The prompt for response generation in the ablation study: *without $R_a^*$*:

```
[Task] You are a medical assistant.
Your task is to understand the ongoing conversation
and continue the medical inquiry in English.

[Guidelines]
- Each response must contain exactly one clear and concise medical question
  with 2 to 3 answer choices.
- Do not repeat any previous question.
- Your response must be a single sentence.
```

The prompt for the reward grading of $\Omega$ (format score) and $R_a$ (content score):

```
[Task] You are an evaluation assistant.
The user will provide a dialogue history between a doctor and a patient.
You must analyze the dialogue and evaluate the doctor's last message.

[Grading Policy]
Format Score:
- 1.0: The doctor's last message contains exactly **one question**.
- 0.5: The doctor's last message contains **two questions**.
- 0.0: The doctor's last message contains **three or more questions**.

Content Score:
- 1.0: The question(s) **directly ask about** any item
    in the Reference Information.
- 0.5: The question(s) are **highly relevant** to,
    but not directly asking about, any item in the [Reference Information].
- 0.0: The question(s) are **irrelevant** to all items
    in the Reference Information.
```

```
[Reference Information]
{The extracted information is inserted here.}

[Output Format]
<think>
Explain your reasoning for the format and content scores
clearly and concisely.</think>
<format_score>
Insert only the format score as a float (e.g., 1.0, 0.5, 0.0)
</format_score>
<content_score>
Insert only the content score as a float (e.g., 1.0, 0.5, 0.0)
</content_score>

[Important]
- Output **exactly** the three tags shown above.
- Do **not** include any additional text, explanation,
    or formatting outside the tags.
- Scores must be based **only** on the doctor's **last message**
    and the provided Reference Information.
- Ensure clarity and precision in your evaluation reasoning
    within the `<think>` tag.
```

## H EVALUATION ON GENERAL CAPABILITIES BENCHMARKS

To assess the impact of our fine-tuning process on the models' general abilities, we conducted evaluations across a range of public benchmarks focusing on domain capability (MedJourney (Wu et al., 2024), MedAgents (Tang et al., 2025)), safety (MedSafety (Han et al., 2024), MedHallu (Pandit et al., 2025), Flames (Huang et al., 2023)), instruction following (IFEval (Zhou et al., 2023), InfoBench (Qin et al., 2024), StructFlow (Li et al., 2025a)), and inference performance (EvalScope Perf (ModelScope Team, 2024)). Fig. 5 presents the full results for the 7B and 32B models, respectively.

Our findings indicate that the specialized training for proactive dialogue does not harm the model's core competencies. Performance on domain-specific tasks (MedAgents, MedJourney) and instruction-following benchmarks (IFEval, StructFlow) remains stable or slightly improves. We observe minor trade-offs in safety-related metrics, such as a decrease in hallucination detection on MedHallu for the 7B model, which warrants careful monitoring in real-world applications. Overall, the *Learn-to-Ask* framework successfully imbues the model with a new and complex skill while largely preserving its foundational capabilities.

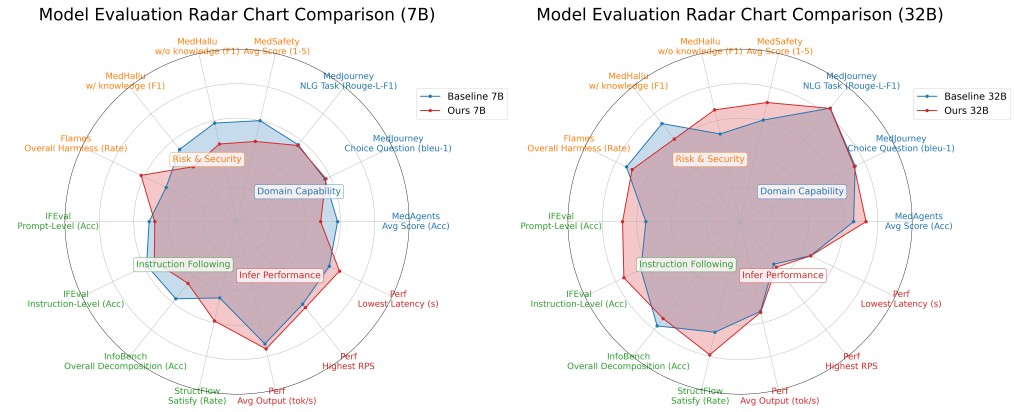

Figure 5: The evaluation results on general capabilities benchmarks on our models with 7B and 32B parameters.

# I DETAILED ANALYSIS OF AUTO-PROMPT

The Auto-Prompt variant automatically calibrates the policy sampler prompt, aiming to improve the quality of exploration during Reinforcement Finetuning (RFT). Tab. 3 shows the results compared to our main method which uses a fixed, manually-crafted sampler prompt. The optimized prompt is obtained from 30 iterations of automated prompt optimization pipeline mentioned in Section E, where the average total reward on a calibration dataset with 100 samples is increased from 2.69 to 3.07.

Table 3: Comparison of the models trained with the original prompt and optimized prompt.

| Method | WA | WA-GH | WC | WS | AA | FC | TR |
|---|---|---|---|---|---|---|---|
| | | **Results on 7B Models** | | | | | |
| Base | 0.501 | 0.132 | 0.975 | 0.155 | 0.751 | 0.629 | 2.174 |
| Original | **0.665** | **0.413** | 0.944 | **0.926** | **0.939** | **0.915** | **3.272** |
| Optimized | 0.641 | 0.399 | **0.949** | 0.910 | 0.938 | 0.894 | 3.214 |
| | | **Results on 32B Models** | | | | | |
| Base | 0.503 | 0.134 | 0.915 | 0.521 | 0.807 | 0.670 | 2.431 |
| Original | **0.640** | 0.365 | **0.933** | 0.877 | 0.918 | 0.880 | 3.145 |
| Optimized | 0.634 | **0.366** | 0.925 | **0.916** | **0.923** | **0.889** | **3.166** |

The optimized prompt is:

```
[System] You are a health consultant.
Your role is to comprehend the ongoing conversation and
pose a medical question in English.

[Guidelines]
- Ensure each reply includes precisely one clear medical inquiry
  with 3 or 4 response choices.
- Avoid repeating any earlier questions.
- Restrict your answer to a single sentence.
- Once sufficient data is collected for a drug suggestion,
  simply output: <stop />
```

On this academic dataset, the performance gains from Auto-Prompt are marginal. We hypothesize this is for two reasons. First, the task in `RealMedConv` is relatively focused, and a simple, well-crafted manual prompt can already generate a high-quality candidate space. Second, larger models like the 32B may be less sensitive to minor variations in the sampler prompt compared to the 7B model.

In contrast, in our large-scale production environment—where the dataset is over 100x larger, covers 10x more medical conditions, and the prompt must incorporate complex business rules—manual prompt engineering becomes intractable. In that setting, the systematic, automated approach of Auto-Prompt is not just beneficial but essential for achieving robust performance and maintaining the system over time. This highlights a key takeaway: the value of certain methodological components may only become fully apparent at industrial scale.

