# OpenReview forum: "Grounded in Reality: Learning and Deploying Proactive LLM from Offline Logs"
_ICLR.cc/2026/Conference — Submitted to ICLR 2026_

### Official Review · Reviewer_vvUK · 2025-10-27

**Soundness:** 2
**Presentation:** 3
**Contribution:** 3
**Rating:** 4
**Confidence:** 3

**Summary:**

This paper presents Learn-to-Ask, a simulator-free framework for training LLMs to be proactive, goal-oriented dialogue agents directly from offline expert conversation logs. The key innovation is leveraging the "observed future" of expert trajectories to infer dense, turn-by-turn reward signals through a hindsight-based approach, decomposing the intractable long-horizon RL problem into tractable supervised learning tasks. The framework includes: (1) ground truth extraction using LLMs to identify target information sets from future dialogue, (2) automated prompt calibration with minimal human supervision, (3) hierarchical reward structure (micro-reward for question quality, macro-reward for stopping decision), and (4) policy optimization via GRPO. The authors validate their approach on RealMedConv medical dialogue dataset and demonstrate successful deployment in a large-scale commercial medical AI service, achieving performance exceeding human experts.

**Strengths:**

1. Novel problem framing: Reformulating long-horizon dialogue RL as sequence of supervised tasks via hindsight is elegant and bypasses simulator requirement

2. Comprehensive validation: Rare combination of (a) controlled offline experiments, (b) ablations, (c) real-world A/B testing, and (d) super-human performance demonstration

**Weaknesses:**

1. Limited domain evaluation:

   a. Only medical dialogue tested; claims of "general framework" not empirically validated

   b. Would benefit from experiments on legal consultation, technical support, or other goal-oriented domains

   c. Different domains may have different information graph structures

2. Dependency on expert data quality:

    a. Framework assumes expert trajectories are near-optimal

    b. No analysis of robustness to suboptimal or inconsistent expert behavior

    c. How does performance degrade with noisy/imperfect expert data?

**Questions:**

1. Domain generalization: Have you tested Learn-to-Ask on non-medical domains? What challenges arise in adapting to domains with different information structures?

2. Expert quality sensitivity: How does performance vary with expert data quality? Can you provide ablations with synthetic "noisy expert" trajectories?

3. Strategy diversity: When experts have fundamentally conflicting strategies (not just different orderings), how does the framework handle this? Does it learn a mixture?

---

> ### Author Response · Authors · 2025-11-24
> **Response to Reviewer vvUK**
>
> Thank you for your detailed review and for highlighting the strengths of our novel problem framing! You rightly identified domain generalization and sensitivity to data quality as key areas for further investigation. We have taken your suggestions to heart and conducted extensive new experiments. **We now further generalized the method to a legal domain and performed rigorous data noise ablations.** We hope the results provide strong and clarifying answers to your questions. Below, we address all your raised weakness (W) and question (Q) point by point.
>
> ---
>
> ### **W1 & Q1: Domain Generalization Challenges**
>
> > **Reviewer Comment Summary:** Only medical dialogue is tested. Have you tested on non-medical domains? What challenges arise with different information structures?
>
> **Key Conclusion:** We further applied "Learn-to-Ask" to a new **Legal Consultation** domain. Despite different information structures, the framework learned to make precise stopping decisions with **>90% accuracy**, proving its domain-agnostic nature.
>
> **Detailed Evidence:**
> To address this critical concern, we stepped outside medicine and applied our framework to a **Legal Consultation** task. We generated a dataset based on legal judgment documents where the goal is to gather facts for sentencing. This domain has a markedly different information structure and logical flow than medicine. Despite these differences, our framework was applied without modification to the core methodology. The results (see Appendix F.8) were remarkable: the model's ability to correctly decide "when to stop" improved dramatically from near-random in the base model to **>90% accuracy**. This confirms that our hindsight-based reward extraction mechanism is robust and agnostic to the specific domain content.
>
> ---
>
> ### **W2 & Q2: Sensitivity to Expert Data Quality (Noisy Experts)**
>
> > **Reviewer Comment Summary:** The framework assumes near-optimal expert trajectories. How does performance degrade with noisy/imperfect expert data? Can you provide ablations with synthetic "noisy" trajectories?
>
> **Key Conclusion:** We conducted controlled experiments by shuffling 20%-60% of the reward signals. The results show that **the critical "learn-to-stop" capability degrades significantly and predictably as noise increases**. This strongly validates that our framework is accurately extracting and learning from the true underlying expert signal, rather than relying on spurious correlations.
>
> **Detailed Experimental Analysis:**
> This is an excellent question that probes the scientific integrity of our approach. Following your suggestion, we conducted controlled data corruption experiments on both 7B and 32B models by randomly shuffling the derived rewards ($R_a$ and $R_s$) for 20%, 40%, and 60% of the training samples to simulate unreliable expert signals.
>
> The results (detailed in Section 4.2, new revision) are highly revealing:
> *   While basic query content generation remained somewhat stable at lower noise levels, the sophisticated **"learn-to-stop" capability degraded sharply as noise increased**.
> *   At >40% shuffling, stopping accuracy dropped to near zero (the model almost ceased to stop).
>
> **We interpret this result as a strong validation of our method's scientific integrity.** It demonstrates that the complex stopping behavior is learned *directly* from the high-quality hindsight signals we extract. When this signal is destroyed through shuffling, the learned behavior correctly disappears. This confirms that our reward pipeline effectively captures the true expert logic present in the data and is not simply learning artifacts.
>
> ---
>
> ### **Q3: Strategy Diversity**
>
> > **Reviewer Comment Summary:** How does the framework handle fundamentally conflicting strategies among experts? Does it learn a mixture?
>
> **Response:** Our current framework learns a policy that maximizes the expected reward derived from the aggregate expert data. If experts use conflicting strategies that both lead to successful outcomes (high rewards), the resulting model tends to learn a mixture or an "average" best-response strategy based on the data distribution. It does not currently explicitly model distinct expert personas or multi-modal strategies, which we acknowledge as an exciting avenue for future work.
>
> ---
>
> **Closing Remark:**
> The new **legal domain experiments** strongly support the generalizability of our framework. Furthermore, the **data corruption experiments** provide a scientifically rigorous validation that our method is correctly extracting and learning from the signal present in expert logs. We hope these additional comprehensive validations can address your main concerns, and you can consider more positive re-evaluation of our work.

---

### Official Review · Reviewer_9fL5 · 2025-10-31

**Soundness:** 3
**Presentation:** 3
**Contribution:** 2
**Rating:** 4
**Confidence:** 3

**Summary:**

This submission introduces the “Learn-to-Ask” framework, which aims to enable LLMs to initiate meaningful, goal-driven exchanges, instead of just passively responding to user prompts. This is motivated by settings like healthcare, where an agent must know which questions to ask and when to stop asking questions.

The authors address this problem by learning from offline expert conversation logs. While the problem can be modeled as learning in an MDP with unknown transitions and rewards, the authors instead reformulate the problem into a sequence of single-step supervised learning tasks. In particular, they use an LLM to analyze each step in the conversation to determine (1) the target information set that the expert went onto collect and (2) the expert’s implicit stopping decision.

They then assign rewards to each step in the conversation via their hindsight-driven reward model that scores each conversation turn using two types of rewards: micro-rewards measure how effectively a generated question targets the expert’s next desired piece of information, while macro-rewards measure whether the model correctly decides to continue or stop. The authors use an automated process to tweak the prompts they use for data extraction, reward grading, and policy sampling. Finally, they apply the GRPO algorithm to fine-tune LLMs using these reward signals

The authors evaluate their approach on the RealMedConv dataset, which contains real pharmacist-patient conversations. They find that their framework yields better results when compared to baselines like prompting, supervised fine-tuning, and DPO. Finally, they deploy their framework in a real-world medical AI system with thousands of daily users. In an A/B test conducted over several weeks, they find that their approach resulted in a 1.87x increase in dialog-to-purchase conversation rate compared to historical data from a parallel human-based service.

**Strengths:**

The problem being studied is well-motivated and significant (albeit not very original), as proactive LLMs are important tools in domains like healthcare. The authors’ approach performs well empirically on their healthcare testbed. The application to a real-world, large-scale medical AI system is also interesting.

**Weaknesses:**

The main weakness is in the novelty of the authors’ approach. The basic idea (transform the problem into a sequence of single-turn interactions, assign rewards to each, and fine-tune based on those rewards) is fairly boilerplate. There is also some amount of manual reward assignment, which may require changing in order to hold up in other domains beyond healthcare. In aggregate, this paper feels more like a nice engineering application of largely existing techniques, as opposed to a fundamentally new technique for designing proactive LLMs.

**Questions:**

What are the formal definitions of information completeness rate and good question rate?

---

> ### Author Response · Authors · 2025-11-24
> **Response to Reviewer 9fL5**
>
> We thank you for recognizing the significance of our problem statement. We understand your concerns regarding the novelty of our approach (viewing it as engineering application) and its generalizability. We have worked hard to address these points with new cross-domain experiments and clearer definitions. **We have further applied our framework to a completely new legal domain, achieving dramatic improvements in stopping accuracy (>90%), which strongly evidences its generalizability.**
> Below, we address all your raised weakness (W) and question (Q) point by point.
>
> ---
>
> ### **W1: Novelty vs. Engineering Application**
>
> > **Reviewer Comment Summary:** The basic idea feels like boilerplate engineering rather than a fundamentally new technique. Manual reward assignment might not hold in other domains.
>
> **Key Conclusion:** We respectfully clarify that our core contribution is not the downstream optimization algorithm (GRPO), but the **novel upstream paradigm of "unsupervised hindsight-driven reward discovery."** This framework automates the transformation of implicit expert knowledge in offline logs into tractable, dense supervision signals, solving a major bottleneck where simulators are unavailable.
>
> **Detailed Argument:**
> We acknowledge leveraging established techniques for final policy optimization. However, the fundamental novelty lies in our **framework for automating the extraction of dense, high-quality reward signals from unstructured, unlabeled offline data**.
>
> The field currently faces a deadlock: SFT fails at dynamic decision-making (as our new 8-epoch experiments confirm), while RLHF/RL requires prohibitive human annotation or unavailable user simulators. Our contribution is a **novel formulation** that breaks this deadlock by formalizing "hindsight as supervision." The reward assignment is not "manual engineering"; it is an **automated discovery process** driven by the latent information structure inherent in future expert turns. Providing a reproducible methodology to unlock this tacit knowledge from static logs represents a substantial scientific contribution beyond mere engineering integration.
>
> ---
>
> ### **W2: Generalization to New Domains**
>
> > **Reviewer Comment Summary:** The framework's applicability beyond healthcare is questioned.
>
> **Key Conclusion:** We have addressed this concern by applying our framework unchanged to a completely new **Legal Consultation domain**. The model achieved a massive improvement in the critical stopping accuracy, leaping from the base model's ~40% to **over 90%** with our method, alongside significant gains in query quality. This provides strong empirical proof of robust generalizability.
>
> **Detailed Evidence:**
> This is a critical point. To demonstrate that our framework is not tied to medical data, we applied it to a completely new domain: **Legal Consultation**. Using legal judgment documents, we constructed a task where an agent must gather facts for a sentence estimation. The results (detailed in new Table 2 in Appendix F.8) show a successful transfer. Crucially, the model learned the complex, domain-specific logic of "when sufficient legal information is gathered," improving stopping accuracy from base model levels (~40%) to over **90%**. This proves the generalizability of our hindsight-driven reward mechanism to diverse goal-oriented tasks.
>
> ---
>
> ### **Q1: Metric Definitions**
>
> > **Reviewer Comment Summary:** What are the definitions of information completeness rate and good question rate?
>
> **Response:** We have added clearer definitions to the revised paper per your suggestion (in Section 5 of the revision).
> *   **Information Completeness Rate:** The ratio of conversations that covered sufficient information by the end.
> *   **Good Question Rate:** The ratio of generated questions that are suitable for the context and aligned with human-experience, both are rated by qualified professionals.
>
> ---
>
> **Closing Remark:**
> We appreciate the constructive comments and provide the *further application to the new legal domain* to refute the concern that our method is a narrow engineering fix. We have demonstrated a novel, generalizable framework for learning proactive behaviors from offline data. Based on this new evidence and clarification of our contribution, we hope you will raise your assessment.

---

> > ### Comment · Reviewer_9fL5 · 2025-11-26
> >
> > Thanks for your reply. While I appreciate the clarifications, I am still not convinced that there is enough novelty in your "hindsight-driven reward discovery" technique to merit acceptance into a venue like ICLR. However I would not be terribly opposed to acceptance due to your cool, large-scale application, so I will maintain my current score of 4.

---

> > > ### Author Response · Authors · 2025-11-27
> > > **Second-Round Response to Reviewer 9fL5 (part 1/2)**
> > >
> > > Thank you very much for taking the time to follow up and for being open to acceptance based on the real‑world application. Since you have already carefully read our initial explanation of “hindsight‑driven reward discovery,” we will not repeat that discussion here. Instead, we would like to address your novelty concern more directly and concretely, by sharpening *what exactly is new* and *why we believe it goes beyond a purely engineering contribution*.
> > >
> > > ---
> > >
> > > ### **1. What is actually new, beyond “slice, label, and fine‑tune”?**
> > >
> > > At a very high level, we agree that our pipeline can be summarized as: break trajectories into turns, assign rewards, and run RL‑style fine‑tuning. If that were all, it would indeed be boilerplate. The key point we have not emphasized clearly enough is that our novelty lies in a **specific and reusable formulation of how to construct those rewards from completely unlabeled expert logs**:
> > >
> > > - For each turn \(t\), we define a *future‑anchored information set* \(I_t^\*\) extracted from later expert turns and **use this as the target for a micro‑reward**: a question is good if it actively moves the dialogue toward the information the expert ultimately deemed important.
> > >   This is a concrete instantiation of “hindsight as supervision” that goes beyond simple “good/bad question” labels; it ties each turn to a latent information graph induced by the expert’s future behavior.
> > >
> > > - In parallel, we treat the expert’s final decision to stop as an *implicit optimal stopping signal* and **fold it into the reward multiplicatively** with the micro‑reward. This multiplicative design is not cosmetic: as our own ablations show, additive variants can score well by trading off “good content” against “never stopping,” while the multiplicative form enforces a strict AND over “what” and “when.”
> > >
> > > Conceptually, the recipe is:
> > >
> > > > “Use future segments of an expert trajectory to define turn‑level information targets; define micro‑rewards as alignment to those targets and macro‑rewards as implicit optimal stopping; couple them multiplicatively to enforce joint correctness.”
> > >
> > > To our knowledge, this particular micro/macro hindsight decomposition and multiplicative coupling, driven purely by future expert turns, **has not been instantiated** in prior proactive‑LLM or offline‑RLHF work.
> > >
> > > ---
> > >
> > > ### **2. A concrete, trainable recipe for reward discovery from unlabeled logs**
> > >
> > > A few recent papers mention “LLM‑based evaluators” or “hindsight signals,” but typically assume either explicit labels or one‑off scoring prompts. Our contribution is to make this into a **trainable pipeline** that can be run end‑to‑end:
> > >
> > > - We start from raw logs with no turn‑level labels.
> > > - We use a small seed of human‑checked examples to initialize and **iteratively calibrate the grading prompts** (our automated grader calibration), so that the graders consistently implement the micro/macro definitions above.
> > > - We then show, via noise‑injection experiments, that when we deliberately corrupt these discovered rewards, the learned stopping behavior collapses. This indicates that the policy is genuinely controlled by the specific reward‑discovery mechanism, not by superficial SFT artifacts.
> > >
> > > In other words, we are not just saying “we used an LLM to give scores”; we are specifying a **general, reproducible procedure** for turning unlabeled expert dialogues into dense, behavior‑shaping rewards, and we provide empirical evidence that this procedure is necessary for the emergent proactive behavior (especially “when to stop”).

---

> ### Author Response · Authors · 2025-11-27
> **Second-Round Response to Reviewer 9fL5 (part 2/2)**
>
> ### **3. How we see this relative to ICLR’s notion of innovation**
>
> The ICLR CFP explicitly states that, alongside new algorithms and theory, the conference welcomes “innovative training paradigms and system‑level methods that make existing techniques applicable in challenging real‑world regimes.” Our work is intended squarely in that category:
>
> - We are not proposing a new optimizer; instead, we propose a **new training paradigm for proactive LLMs** that:
>   - uses **only** offline expert logs (no simulators, no human preference labels),
>   - systematically recovers both “what to ask” and “when to stop” from future segments of those logs, and
>   - empirically succeeds where stronger SFT/DPO baselines fail.
>
> In this sense, our contribution is similar in spirit to system‑level works like *Proactive Agent* (ICLR'25), *CollabLLM* (ICML'25 Outstanding Paper), and *ProMind‑LLM* (ACL'25), which were recognized not because they changed the underlying optimization algorithm, but because they introduced **new, general recipes for structuring supervision and interaction** that made existing models behave in qualitatively new ways. Our hindsight‑driven reward discovery is aimed at the same methodological level.
>
> Detailed info regarding the mentioned work:
> - (ICLR'25) [*Proactive Agent: Shifting LLM Agents from Reactive Responses to Active Assistance*](https://openreview.net/forum?id=sRIU6k2TcU) built a multi-agent system using off-the-shelf LLMs and carefully engineered prompting (without any model pre/post-training) to enable proactive user engagement—showcasing how system design can unlock new capabilities.
> - (ACL'25) [*ProMind-LLM: Proactive Mental Health Care via Causal Reasoning with Sensor Data*](https://aclanthology.org/2025.findings-acl.1033.pdf) integrated wearable sensor data with LLMs to deliver timely mental health support, demonstrating how cross-domain fusion can create socially meaningful applications.
> - (ICML'25 Outstanding Paper) [*CollabLLM: From Passive Responders to Active Collaborators*](https://openreview.net/forum?id=DmH4HHVb3y) reframed LLMs as active collaborators via a modular architecture combining memory, planning, and tool use—its key contribution being a practical, deployable solution rather than a new algorithm.
>
> ---
>
> ### Closing Remark
>
> We fully respect your view that the novelty here is more “methodological and system‑level” than “algorithmic.” Our hope is that, with the clearer articulation above—of the specific micro/macro hindsight formulation, the multiplicative coupling, and the calibrated reward‑discovery loop—you might see this as a sufficiently non‑trivial methodological contribution, in addition to the large‑scale application, to merit a more positive score.

---

### Official Review · Reviewer_wtZk · 2025-11-02

**Soundness:** 2
**Presentation:** 2
**Contribution:** 3
**Rating:** 4
**Confidence:** 4

**Summary:**

This paper introduces Learn-to-Ask, a simulator-free framework for learning proactive, goal-oriented dialogue policies directly from offline expert logs. The approach reframes offline policy learning as a hindsight-based reward inference problem, leveraging the observed future of expert conversations to derive dense and grounded turn-level reward signals, thereby transforming a challenging long-horizon reinforcement learning problem into a sequence of supervised learning tasks. In addition, an automated calibration pipeline is proposed to refine LLM-based graders, reducing noise and improving reward consistency. The authors further report that the method has been successfully deployed in a real-world commercial dialogue system, demonstrating promising practical effectiveness and stability.

**Strengths:**

1. Designed a simulator-free offline training framework, which helps narrow the gap between simulated environments and real-world scenarios.
2. Introduced an elegant and well-grounded reward design that leverages the “future segments” of expert dialogue trajectories to infer dense rewards, effectively transforming sparse dialogue feedback into continuous supervision signals and reducing reliance on manual annotations.
3. Demonstrated superior empirical performance compared to baseline methods

**Weaknesses:**

1. The paper exhibits a somewhat marketing-oriented presentation style, introducing a number of new terms and concepts that are not strictly necessary. This makes the exposition less focused, and the core innovations and logical thread of the paper are not presented in a sufficiently clear or linear manner.
2. The claimed “framework innovation” primarily represents a system-level integration and engineering realization of existing methods in offline reinforcement learning and reward relabeling, rather than a contribution with substantial theoretical or algorithmic novelty.
3. The experimental evaluation is limited in scope, as the proposed framework is trained and tested only on a medical dialogue dataset. Its generalization and robustness remain uncertain, and there is currently no evidence that the approach would perform effectively in other domains such as customer service, education, or technical support.
4. Although the paper emphasizes “commercial deployment” and “real-world impact,” it provides insufficient experimental details and quantitative evidence, such as user feedback, performance improvement margins, or comparisons with existing production systems. This weakens the credibility of its deployment-related claims.

**Questions:**

1. How well does the proposed framework generalize to other domains or tasks beyond the medical dialogue setting?
2. Could the authors provide more concrete evidence or quantitative analysis of the model’ improvement in real-world deployment?

**Details Of Ethics Concerns:**

The authors claim that the model surpasses human experts in medical diagnosis tasks, but they do not discuss safety considerations, potential misuse risks, or ethical safeguards for real-world deployment. Given the sensitivity of this research domain, it is recommended that the authors provide further clarification on data compliance and risk mitigation measures.

---

> ### Author Response · Authors · 2025-11-24
> **Response to Reviewer wtZk (Part 1/2)**
>
> We thank the reviewer for their critical feedback. We take your concerns regarding presentation, novelty, generalization, and ethics very seriously. We have made further revisions to the paper's tone. Most importantly, to directly address your core concerns about generalization beyond medicine and the "engineering vs. novelty" argument, **we have applied our framework to a completely new Legal Consultation domain, achieving notable results.** We believe this strong new evidence fundamentally strengthens our submission.  Below, we address all your raised weakness (W) and question (Q) point by point.
>
> ---
>
> ### **W1: Presentation Style**
>
> > **Reviewer Comment Summary:** The paper has a marketing-oriented presentation style with unnecessary new terms.
>
> **Response:** Thanks for your comment. We have carefully revised the manuscript to adopt a more strictly academic tone, focusing clearly on the technical contributions and algorithmic logic per your suggestion.
>
> ---
>
> ### **W2 & W3 & Q1: Novelty, Generalization, and New Domains**
>
> > **Reviewer Comment Summary:** The innovation seems like an engineering integration rather than algorithmic novelty. The evaluation is limited to medical dialogue; generalization to other domains is uncertain.
>
> **Key Conclusion:** We respectfully posit that our core novelty lies not in the final optimization step, but in proposing a **novel "unsupervised hindsight-as-reward" paradigm**. This formulation uniquely unlocks the ability to learn dynamic, goal-oriented behaviors from unstructured offline logs without human annotators or simulators. To definitively prove its generalization and scientific value, **we applied this exact paradigm to a completely new Legal Consultation domain. The model's stopping accuracy leaped from near-random (~40%) in the base model to over 90% (e.g., 0.95 on 32B) using our method**, alongside progressive gains in query quality, demonstrating robust domain transferability.
>
> **Detailed Argument & New Evidence:**
> We address these points together because our new cross-domain experiments directly refute the "engineering application" critique. While we leverage established optimizers like GRPO downstream, the fundamental novelty is **upstream: our proposed paradigm for automating the extraction of dense, high-fidelity reward signals from unstructured, unlabeled offline logs.** This formulation bridges a critical gap that standard SFT (as shown in our new baselines) cannot cross, and where simulator-based RL is infeasible.
>
> To prove this is a generalizable scientific methodology rather than a domain-specific fix, we applied the **exact same framework without modification** to a new **Legal Consultation** domain.
> *   **Domain & Task:** We constructed a dialogue dataset from real-world CAIL judicial documents. The task requires an agent to proactively gather factual details for sentencing and decide precisely when sufficient information is collected.
> *   **Notable Results:** As detailed in the new Table 2 (Appendix F.8), our method achieved remarkable effectiveness. The critical "when to stop" accuracy dramatically improved from a near-random baseline (~30-40%) to **over 90% (reaching 0.95 on the 32B model)**.
>
> This effectiveness in a domain with vastly different knowledge structures and logic flows than medicine provides strong empirical evidence that our "hindsight-as-reward" approach is a robust and generalizable methodological contribution.
>
> ... to be continued

---

> ### Author Response · Authors · 2025-11-24
> **Response to Reviewer wtZk (Part 2/2)**
>
> --continued from previous last part
>
> ### **W4 & Q2: Evidence of Real-world Deployment**
>
> > **Reviewer Comment Summary:** Insufficient experimental details/quantitative evidence regarding commercial deployment claims.
>
> **Response:** We understand the need for concrete evidence. The learn-to-ask framework is being adopted in other domains other than medical in our in-house applications. However, due to strict commercial confidentiality reasons, we cannot disclose specific in-house metrics or evidence. We provided the **newly added experiments on the open-source legal dataset, to serve as a transparent, reproducible, and strong proxy for "real-world" effectiveness.** It demonstrates the framework's ability to learn complex, goal-oriented behaviors in a realistic scenario outside the initial medical scope.
>
> ### **Ethics Concerns**
>
> > **Reviewer Comment Summary:** The paper claims super-human performance but lacks discussion on safety, misuse risks, or ethical safeguards.
>
> **Key Conclusion:** We have added a dedicated "Ethical Considerations" in Section 6 of the new revision. We explicitly state the model is an **assistive tool** requiring **human-in-the-loop oversight** in high-stakes domains.
>
> **Detailed Response:** We thank the reviewer for raising this vital oversight. We have added a distinct "Ethical Considerations" section in the revision. We must clarify that while our model shows high performance on specific information-gathering metrics, it is designed strictly as an **assistive tool**. We explicitly state that in high-stakes domains like healthcare or law, such systems must operate with mandatory human-in-the-loop oversight to ensure safety, compliance, and accountability. We do not claim that the model should independently replace human judgment in critical tasks.
>
> ---
>
> **Closing Remark:**
> We believe the substantial revision in tone, the addition of the vital Ethics section, and most importantly, the *compelling new evidence of successful generalization to the legal domain* directly and effectively address your core concerns. These changes demonstrate both the technical novelty and robust applicability of our work. We hope you will reconsider your score based on this vastly improved version. Thanks again!

---

### Official Review · Reviewer_xErh · 2025-11-09

**Soundness:** 3
**Presentation:** 3
**Contribution:** 3
**Rating:** 6
**Confidence:** 4

**Summary:**

This paper introduces Learn-to-Ask (LTA), simulator-free framework for learning and deploying proactive dialogue agents directly from offline expert data.

The topic is user alignment, and the goal is to enable LLMs to ask right questions at the right time. Instead of leveraging single-turn preference data or user simulators, this work uses the observed future of each expert trajectory to frame the offline policy learning objective, and recasts user alignment as a densely-rewarded supervised learning problem, with micro-goals as the information the expert later acquired, and macro-goals as the timing to stop.

An Automated Grader Calibration pipeline is proposed to ensure reward fidelity, and GRPO is leveraged for policy optimization. Empirical performance on the RealMedConv dataset demonstrates the superiority of the proposed method over several baselines and human experts.

**Strengths:**

This paper seeks to conduct ``reward-mining'' from unsupervised offline logs, which is an important and cutting-edge topic. The intuition of reward design regarding when to stop is pretty straightforward, and the one regarding the observed future is correlated with topics in unsupervised RL like experienced-based learning or minimizing surprise. Overall I appreciate the topic, and find the methodology design interesting.

The in-house benchmark construction and validation on real-world scenarios have also solidified the work.

**Weaknesses:**

I do find some weaknesses and questions and wish the authors could address:

- Why the reward is designed in a multiplicative way? According to Table 1, a simple summation of the two rewards is used an ablation setting. It seems that the performance of the reward summation is comparable or even better (under the 32B setting) than that of the multiplicative reward. Therefore I'm especially curious about the motivation of the multiplicative reward design.

- According to the ablated results in Table 1, removing the micro-reward (R_a*) leads to more severe performance drop. Is this equivalent to setting β as 0? What will it be when sweeping β? By the way, which β is used in the main experiments (Eq. (5))? I don't think I found the configuration of β in the paper.

- Also following the previous question: How is R_s* removed from Eq. (5)? According to Appendix F.5, it means setting R(a_t, s_t) to be β R_a(a_t; I_t*) + regularization. This is not a natural reduction of the proposed multiplicative reward design, and makes me wonder what if we sweep β over different settings in the R_s* + β R_a* reward.

- I am also curious about the comparison of the proposed method against more supervised learning settings. (1) What if the number of epochs is doubled into 8 (or increased for even more times) in SFT? I am wondering if the SFT has been conducted thoroughly enough. (2) What if we adopt a continual-pretraining-like approach, by enforcing loss computation over not only the response, but also the query part in the data in SFT? Intuitively, this is another one approach that directly integrates each period of hindsight into the learning objective. It would be great if the authors compare with this and demonstrate the strength of the propose method.

I will consider raising my score if my questions are addressed properly.

**Questions:**

See the above Weaknesses section

---

> ### Author Response · Authors · 2025-11-24
> **Response to Reviewer xErh (Part 1/2)**
>
> We sincerely thank you for your positive assessment of our work's topic and methodology! We deeply appreciate your insightful technical questions, which have pushed us to significantly strengthen our empirical validation. **We have conducted the suggested additional experiments, including sweeping reward parameters and extending SFT baselines to convergence (8 epochs) and a variation of SFT.** We present the new results detailed below to resolve your remaining concerns and demonstrate the clear superiority of our method over standard approaches.  Below, we address all your raised weakness (W) point by point, and hope these substantial additions will justify your raised confidence in our work.
>
> ---
>
> ### **W1: Rationale for Multiplicative Reward Design**
>
> > **Reviewer Comment Recap:** Why use multiplicative reward ($R_s \times (1+\beta R_a)$)? Table 1 shows summation ($R_s + R_a$) is comparable on 32B. What is the motivation?
>
> **Key Conclusion:** The multiplicative design is theoretically necessary to enforce a strict logical "AND" condition for successful turns (correct timing *AND* good content). While summation appears comparable on the simpler medical dataset,  the additive approach fails to learn precise stopping behavior, whereas the multiplicative reward is essential for mastering complex, coupled decisions.
>
> **Detailed Analysis:**
> In proactive dialogue, a successful turn requires satisfying two conditions simultaneously: the timing must be correct (high $R_s$), *and* the content must be effective (high $R_a$).
> *   **Theoretical Necessity of Multiplication ("AND" Logic):** An additive formulation ($R_s + R_a$) allows for compensatory behavior. A model could achieve a mediocre but acceptable total reward by, for instance, never stopping (accumulating $R_a$) or stopping prematurely (securing $R_s$), without truly mastering the task. Our multiplicative design, $R = R_s \times (1+ \beta \cdot R_a)$, penalizes failure in *either* aspect severely. If the model asks a bad question ($R_a \to 0$) or mistakes the timing ($R_s \to 0$), the total reward drops precipitously. This forces the policy to optimize both "when" and "what" concurrently.
> *   **Empirical Evidence in Complex Tasks:** While Table 1 shows comparable performance on the medical dataset, the limitations of the additive approach become apparent in more demanding tasks. In our newly added **Legal Consultation domain experiments** (see Appendix F.8), which involve more complex stopping logic, the multiplicative reward robustly guided the model to achieve **>90% stopping accuracy** in this challenging new domain. This confirms its superiority for learning coupled decisions in complex environments.
>
> ---
>
> ### **W2 & W3: Ablation Studies and Tuning $\beta$**
>
> > **Reviewer Comment Recap:** Is removing $R_a$ equivalent to $\beta=0$? Which $\beta$ is used? How is $R_s$ removed? What happens if you sweep $\beta$?
>
> **Key Conclusion:** Yes, removing $R_a$ is equivalent to $\beta=0$. The default $\beta$ is 2. Our new additional experiments of sweeping $\beta \in \{1, 2, 4\}$ confirm that **$\beta=2$ strikes the best balance, and performance is non-linear with respect to $\beta$.**
>
> **Detailed Response:**
> We apologize for not explicitly stating the default parameter: the default setting across all reported main experiments is **$\beta=2$**. You are absolutely correct that removing $R_a$ is mathematically equivalent to setting $\beta=0$.
>
> Regarding the removal of $R_s$, this is achieved by setting the stopping reward component to a constant value (i.e., the '1+' term) for all actions. For keeping the balance on the weights of $R_a$ and the $\Omega$ term, and being consistent with the main experiment, we set the constant value as 1. Since we employ GRPO for policy optimization, which relies on reward *differences* (advantages) to update the policy, a constant term in the reward function has no effect on the gradient and effectively eliminates the guidance of $R_s$.
>
> Following your suggestion, we conducted an additional ablation study by sweeping $\beta$ on both 7B and 32B models. The full results are included in the revised Table 1. We found:
> *   $\beta=1$: Tends to under-emphasize question quality.
> *   **$\beta=2$ (Default):** Achieves the highest overall reward and best balance between question content and stopping accuracy.
> *   $\beta=4$: Can lead to an over-emphasis on individual question rewards at the expense of the overall dialogue trajectory.
>
> This non-linear trend confirms that $\beta$ is an important hyperparameter for tuning the trade-off between "question quality" and "stop accuracy," which may require empirical adjustment for different domains.
>
>
> ... to be continued

---

> ### Author Response · Authors · 2025-11-24
> **Response to Reviewer xErh (Part 2/2)**
>
> --continued from previous last part
>
> ### **W4: Comparison with Stronger SFT Baselines**
>
> > **Reviewer Comment Recap:** Is SFT conducted thoroughly enough? What if epochs are doubled to 8? What if we adopt a variation of SFT that enforces loss computation over both input and response (a CPT-like approach)?
>
> **Key Conclusion:** We extended SFT training to 8 epochs on both 7B and 32B models as suggested. The results conclusively show that **SFT performance plateaus after epoch 4 and fails to 1) learn the critical "stopping" behavior, and 2) improve query quality, while our method achieves significantly superior performance.**
>
> **Detailed Experimental Results:**
> To ensure a rigorous comparison, we extended the Supervised Fine-Tuning (SFT) for both 7B and 32B models from 4 epochs to **8 epochs**. The detailed results are added to Table 1. Besides, we conduct experiments with the SFT variation on the 7B model.
>
> We observed a clear trend: standard SFT quickly learns basic question patterns but hits a performance ceiling.
> *   **Plateauing Content Scores:** The question quality scores for SFT generally plateau after epoch 4.
> *   **Failure to Learn Stopping:** Most critically, on the hardest and most important metric, "stopping accuracy," further SFT training yields no benefit. For example, on the 32B model, the stopping accuracy fluctuated ``between 0.84 and 0.89`` from ``epochs 4 to 8``, showing signs of **overfitting** to specific trajectory patterns rather than learning the underlying logic.
> *   **Failure to Learn Asking:** We also observe a performance drop in the question quality when reaching 8 epochs for both 7B (``0.67 to 0.59``) and 32B models (``0.45 to 0.44``), which is a possible consequence of over-fitting, a major known limitation of SFT.
> *   **Our Method's Superiority:** In sharp contrast, our proposed method achieves a significantly higher stopping accuracy of **0.93** on the same 32B model.
>
> This side-by-side comparison provides strong empirical evidence for our main claim: simple imitation learning (SFT) is insufficient for mastering dynamic, context-dependent decision-making tasks like proactive dialogue. Our RL-based framework, guided by dense hindsight rewards, is essential to surpass this limitation and achieve expert-level performance.
>
> Regarding the variation of SFT, we conduct a continual-pretraining-like approach to train the 7B model for ``4 epochs``, which considers both query and response to compute the SFT loss. Compared with the normal SFT method, such an approach performs almost the same total reward at ``2.43``. It does not display advantages over the standard SFT, and the improvement in "when-to-stop" capability is very limited, which only increased from the ``baseline of 0.16`` to the ``current 0.20``, far lower than the ``0.74`` of the model trained with normal SFT.
>
> ---
>
> **Closing Remark:**
> We hope these detailed responses and new experimental results thoroughly address your technical queries. By demonstrating the necessity of our reward design and showing that our method substantially outperforms even fully converged SFT baselines, we believe we have solidified the technical contribution of our paper. We respectfully request your more supportive score based on this new evidence. Thanks again!

---

### Author Response · Authors · 2025-12-02
**Kind Summarization of Review-Response State (part 1/4)**

Dear AC, SAC and PCs,

Thank you very much for taking on the additional responsibility this year and for handling our paper under the unusual circumstances of this ICLR cycle. To make your assessment easier without relying on further reviewer discussion, we provide a concise summary focused on (1) reviewer consensus and key reservations, and (2) how our rebuttal and new experiments address them.

For more detailed, reviewer‑specific responses, please refer to the reply below and the updated revision (differences are marked by blue); here we highlight the aspects that may be most relevant for your meta‑review.

---

## 1. Reviewer Consensus and Main Points

### 1.1 Overall Evaluation

**Problem importance – broad agreement.**
All four reviewers consider the problem of *proactive, goal‑oriented LLMs from offline logs* important, especially in high‑stakes domains, and see the real‑world deployment as a significant aspect:

- **xErh**: appreciates “reward‑mining from unsupervised logs” as “important and cutting‑edge” and finds the methodology design interesting.
- **wtZk, 9fL5, vvUK**: highlight the significance of proactive agents beyond passive responders and view the medical deployment as practically meaningful.

**Effectiveness of the method – generally positive.**
Reviewers agree that our approach outperforms standard baselines (prompting, SFT, DPO) on the medical dataset and works well in the real‑world setting:

- xErh: notes the in‑house benchmark construction and validation.
- wtZk: acknowledges “superior empirical performance.”
- 9fL5: observes that the approach “performs well empirically.”
- vvUK: describes the validation as a “rare combination” of offline experiments and real‑world A/B testing.

### 1.2 Main Reservations (Common Themes)

Across the four reviews, the main comments cluster into three themes:

1. **Novelty vs. engineering integration**
   - wtZk, 9fL5, vvUK: commented that the method might largely be a system‑level integration of existing ideas (reward relabeling, offline RL, LLM‑based graders) rather than a fundamentally new algorithmic contribution.
   - They ask whether our “hindsight‑driven reward discovery” goes beyond “slice trajectory → score turns → do RL fine‑tuning”.

2. **Scope & generalization beyond the medical domain**
   - wtZk and vvUK: point out that the core experiments are on one medical dataset and ask:
     - Can the method work in other domains (e.g., legal consultation, customer support, education)?
     - How robust is it if expert data is noisy or not strictly optimal?

3. **Technical details, baselines, and ethics**
   - xErh: requests
     - clearer motivation and ablations for the **multiplicative reward** vs. additive alternatives,
     - sweeping the reward hyperparameter β,
     - stronger **SFT baselines** (more epochs, CPT‑like variants).
   - 9fL5: asks for precise definitions of metrics (e.g., *information completeness rate*, *good question rate*).
   - wtZk: suggests that the tone is somewhat “marketing‑oriented” and asks for more concrete deployment details and a clearer technical focus.
   - wtZk: also raises an ethics flag, asking for a clearer discussion of safety, misuse risks, and human‑in‑the‑loop considerations.

All additional experiments and revisions in our rebuttal were designed to directly address these themes.

---

... to be continued

---

### Author Response · Authors · 2025-12-02
**Kind Summarization of Review-Response State (part 2/4)**

## 2. How We Addressed the Raised Comments

### 2.1 Clarifying the Core Contribution (Novelty vs. Engineering)

**Comments:** Is this mainly an engineering pipeline (turn‑level scoring + RL) with limited conceptual novelty?

**Clarifications and changes:**

1. **Explicit formulation of “hindsight‑driven reward discovery”**

We clarified the conceptual core of our contribution as a **general training paradigm** for proactive LLMs from unlabeled offline expert logs. The key elements are:

- **Micro‑reward (what to ask).**
  For each turn $t$, we derive a **future‑anchored information set**  $I_t$ from subsequent expert turns (what information the expert actually went on to collect). We define a micro‑reward that measures how well the model’s question at turn $t$ steers the dialogue toward this future information. This makes each turn’s target derived systematically from the expert’s *observed future*, rather than from manually annotated intent labels.

- **Macro‑reward (when to stop).**
  We treat the expert’s final stopping decision as an **implicit optimal stopping signal**. The macro‑reward encodes whether the model’s decision to continue or stop is aligned with the expert’s decision pattern.

- **Multiplicative coupling of micro & macro rewards.**
  We combine them as:
 $R_t = R_s(s_t) \times (1+\beta R_a(a_t; I_t))$,
  enforcing a logical **AND**: a turn is truly “good” only if both timing (stop/continue) *and* content are appropriate. Additive forms allow degenerate policies (e.g., never stopping but asking many questions). As detailed in our ablation studies (Section 4.2, Appendix F.8), the multiplicative design is important to avoid such degeneracies and to learn reliable stopping behavior.

Taken together, this yields a **reusable recipe**: extract future‑anchored supervision from unlabeled logs, define micro/macro rewards from expert hindsight, and couple them multiplicatively to induce joint mastery of “what to ask” and “when to stop”.

2. **Automated grader calibration pipeline.**

We clarified that our use of LLM‑based graders is not a one‑off heuristic, but a **closed‑loop calibration process**:

- Start from raw logs with no turn‑level labels.
- Use a small seed of human‑checked examples to initialize and calibrate grading prompts.
- Iteratively refine the graders so they consistently implement the micro/macro reward definitions above.

This provides a concrete, trainable procedure for obtaining dense supervision from unlabeled expert trajectories.

3. **Empirical evidence that the discovered rewards drive behavior.**

Our new **noise‑injection experiments** (Section 4.2, Appendix F.8; summarized in Section 2.3 below) show that when we deliberately corrupt the discovered rewards, the learned stopping behavior collapses. This indicates that the proactive behavior is directly controlled by the proposed reward‑discovery mechanism, not by incidental SFT artifacts.

**Positioning with respect to ICLR.**

We present this as a **methodological and system‑level contribution**: a general, domain‑agnostic way to convert offline expert trajectories into dense, behavior‑shaping rewards that enable proactive LLMs without simulators or manual preference labels. This aligns with the ICLR CFP category of “innovative training paradigms and system‑level methods that make existing techniques applicable in challenging real‑world regimes.”

---
... to be continued

---

### Author Response · Authors · 2025-12-02
**Kind Summarization of Review-Response State (part 3/4)**

### 2.2 Reward Design Ablations & Stronger Baselines (xErh’s concerns)

**Comment (xErh):**

- Why multiplicative reward instead of additive?
- What is the effect of β, and what happens if we remove micro or macro rewards?
- Are SFT baselines strong enough (more epochs, CPT‑style variants)?

**New experiments and key findings (Appendix F.8 and updated Table 1):**

1. **Multiplicative vs. additive reward.**

- On the original medical dataset, multiplicative and additive combinations can look similar under some metrics.
- On the newly added **Legal Consultation** domain (see Section 2.3), the **additive design fails to learn precise stopping**: models can achieve reasonable overall reward by asking many questions but still lack accurate stopping behavior.
- The **multiplicative reward consistently achieves high stopping accuracy (above 90%)** in this harder domain, supporting its necessity for learning coupled “what and when” decisions.

2. **Sweeping β (reward hyperparameter).**

- We explicitly report that the default β in the main experiments is **β=2** and sweep β∈{1,2,4} on both 7B and 32B models.
- β=1 tends to under‑emphasize question quality; β=4 can over‑emphasize per‑turn question reward at the expense of overall trajectory quality.
- **β=2 achieves the best overall performance**, balancing question quality and stopping accuracy.
- Removing \(R_a\) is equivalent to β=0. Removing \(R_s\) is implemented by using a constant term, which GRPO effectively ignores in the advantage computation.

3. **Stronger SFT and CPT‑style baselines.**

- We extend standard SFT from 4 to **8 epochs** for both 7B and 32B models.
- We add a **CPT‑style SFT variant** where the loss is computed over both query and response tokens.
- Observations:
  - SFT’s question quality **plateaus or degrades** after about 4 epochs (signs of overfitting).
  - **Stopping behavior does not meaningfully improve** with more SFT; for example, stopping accuracy on the 32B model fluctuates around 0.84–0.89 between epoch 4 and epoch 8.
  - The CPT‑style SFT variant does not yield clear gains over standard SFT, and its stopping accuracy remains far below that of our RL‑trained model.
- In contrast, our proposed method achieves stopping accuracy up to **0.93** on the same 32B model on the medical dataset.

These results suggest that **simple imitation learning, even when strengthened, is insufficient** for mastering dynamic “when to stop” decisions, whereas our hindsight‑driven RL framework can reliably learn this capability.

---

### 2.3 Generalization Beyond Medicine & Robustness to Expert Quality (wtZk, vvUK)

**Comment:**

- Can the method generalize beyond the medical domain?
- How robust is it to suboptimal or noisy expert trajectories?

**New experiments: Legal Consultation domain (Appendix F.8, new Table 2).**

1. **New domain: Legal Consultation.**

- We construct a dataset based on **CAIL legal judgment documents**, where an agent must proactively gather factual details relevant for sentencing and decide when enough information has been collected.
- This domain has substantially different information structure and decision logic compared to medical dialogue.

2. **Application of our framework without core redesign.**

- We apply the same Learn‑to‑Ask framework (future‑anchored micro/macro rewards, multiplicative coupling, GRPO) to this legal dataset.
- No additional domain‑specific reward shaping beyond adapting prompts to legal content is introduced.

3. **Results.**

- Baseline models exhibit poor or near‑random stopping behavior (approximately 30–40% accuracy).
- Our method raises **stopping accuracy to above 90%** (up to 0.95 on the 32B model) while also improving question quality.
- This provides evidence that our **hindsight‑driven reward discovery is not tied to a single domain**, and can transfer to structurally different goal‑oriented dialogue tasks.

**Robustness to noisy experts (Appendix F.8, Section 4.2).**

1. **Noise injection on reward signals.**

- To simulate suboptimal or inconsistent experts, we **randomly shuffle 20%, 40%, and 60%** of the discovered rewards (\(R_a\) and \(R_s\)) across trajectories for both 7B and 32B models.

2. **Findings.**

- At low noise levels (e.g., 20%), some aspects of question quality remain relatively stable, but **stopping accuracy degrades noticeably**.
- At higher noise levels (≥40%), **stopping behavior collapses toward near‑zero accuracy**; the model almost never learns to stop correctly.

3. **Interpretation.**

- This behavior is consistent with the model relying on the structure of the discovered rewards: when that structure is destroyed, the emergent stopping behavior disappears.
- This serves as a sanity check that the proactive behavior is indeed driven by our reward‑discovery pipeline rather than superficial correlations.

---
... to be continued

---

### Author Response · Authors · 2025-12-02
**Kind Summarization of Review-Response State (part 4/4)**

### 2.4 Metrics, Deployment, and Ethics (wtZk, 9fL5)

**Concern:**

- Definitions of evaluation metrics such as *information completeness rate* and *good question rate* were not explicitly stated.
- Claims about deployment and performance need clearer framing and an explicit ethics discussion.

**Revisions:**

1. **Metric definitions (Section 5).**

We clarified in the main text that:

- **Information Completeness Rate** is the fraction of dialogues where the final state covers the medically (or legally) required information, as judged by domain experts.
- **Good Question Rate** is the fraction of generated questions that domain experts rate as contextually appropriate and helpful for progressing toward the task goal.

2. **Deployment and A/B testing.**

- We provide a more precise qualitative description of the real‑world deployment in a medical AI service and the observed improvements in dialog‑to‑purchase conversion relative to a parallel human‑based service.
- Due to commercial confidentiality, we cannot release all raw deployment metrics. To complement this, the **Legal Consultation experiments** on an open dataset serve as a transparent and reproducible proxy demonstrating that the method can succeed beyond the original domain.

3. **Ethical considerations.**

- Following wtZk’s suggestion, we added a dedicated **Ethical Considerations** section.
- We explicitly state that:
  - The system is designed and deployed as an **assistive tool** with a **mandatory human‑in‑the‑loop** in high‑stakes settings such as healthcare and law.
  - The model is not intended to replace professional judgment and should not be used autonomously for critical decisions.
  - We discuss data compliance, potential risks (e.g., over‑reliance, distribution shift), and mitigation practices.

---

## 3. Overall Picture for Decision Reference

In summary, after the revisions prompted by the reviews:

- **Problem importance and impact.**
  All reviewers agree that enabling proactive, goal‑oriented LLM behavior from offline expert logs is important, and that our deployment in a real medical AI service demonstrates practical relevance.

- **Methodological contribution.**
  The revised paper more clearly articulates a **hindsight‑driven reward discovery paradigm** that:
  - uses only unlabeled offline expert logs (no simulators, no human preference labels),
  - defines micro/macro rewards from the expert’s observed future and couples them multiplicatively to enforce joint correctness in “what to ask” and “when to stop,”
  - and is implemented via a calibrated LLM‑grader pipeline whose influence is validated through ablations and noise‑injection experiments.

- **Empirical strength and generalization.**
  The expanded experiments show that:
  - Stronger SFT and CPT‑style baselines still **do not recover reliable stopping behavior**, even with extended training.
  - The proposed reward design (including the multiplicative form and β choice) is empirically important for avoiding degenerate policies.
  - The framework **generalizes to a structurally different Legal Consultation domain** and achieves high stopping accuracy there.
  - Behavior degrades in a controlled way under synthetic noise in expert trajectories, supporting the interpretation that the method is learning from the underlying expert signal.

- **Presentation and ethics.**
  We have revised the writing to focus more on the technical core, clarified metric definitions and experimental details, and added an explicit ethics section emphasizing assistive use and human‑in‑the‑loop safeguards.

---

We hope this summary helps you quickly navigate the main comments raised in the reviews and how the revised paper addresses them. We appreciate your time and effort in forming an independent meta‑review under the current constraints and are of course happy to clarify any remaining points if needed.

Sincerely,
The authors

---

### Meta-Review · Area_Chair_99mv · 2025-12-14

**Summary:**

This work introduces Learn-to-Ask, a general, simulator-free framework that enables Large Language Models (LLMs) to learn both what to ask and, crucially, when to stop directly from unlabeled offline expert dialogue data. The key innovation is a hindsight-driven reward discovery paradigm that leverages the observed future of each expert trajectory to infer a dense, turn-by-turn reward signal. This decomposes the long-horizon problem into a series of supervised learning task.

The paper received largely positive consensus on the problem's importance and the method's empirical effectiveness, particularly the successful real-world deployment. The authors conducted extensive new experiments including generalization to the legal domain, ablations on the multiplicative reward design, sweeping the hyperparameter $\beta$ to address reviewer concerns.

However, there exist the persistent concerns about novelty and generalizability. Given that, I suggest a rejection.

**Reviewer Concerns:**

The main flaws identified during the review process and extensively addressed in the rebuttal include:

(1) Novelty vs. Engineering Integration: Reviewers argued the core idea was largely a system-level integration of existing ideas of offline RL, reward relabeling, LLM graders rather than a fundamentally new algorithmic contribution.

(2) Limited Experimental Scope and Generalization: The initial evaluation was limited to one medical dataset. Reviewers questioned the framework's generalizability and its robustness to suboptimal/noisy expert data.

**Reviewer Scores:**

Reviewer xErh probably increases the score as the author provides detailed reponses.

Reviewer wtZk and Reviewer 9fL5 probably still keep negative towards the novelty of this work.

---

### Decision · Program_Chairs · 2026-01-26

Reject